# Increased ionization supports growth of aerosols into cloud condensation nuclei

H. Svensmark [1], M.B. Enghoff [1], N.J. Shaviv[2] & J. Svensmark [1,3]

Ions produced by cosmic rays have been thought to influence aerosols and clouds. In this study, the effect of ionization on the growth of aerosols into cloud condensation nuclei is investigated theoretically and experimentally. We show that the mass-flux of small ions can constitute an important addition to the growth caused by condensation of neutral molecules. Under atmospheric conditions the growth from ions can constitute several percent of the neutral growth. We performed experimental studies which quantify the effect of ions on the growth of aerosols between nucleation and sizes >20 nm and find good agreement with theory. Ion-induced condensation should be of importance not just in Earth's present day atmosphere for the growth of aerosols into cloud condensation nuclei under pristine marine conditions, but also under elevated atmospheric ionization caused by increased supernova activity.

---

[1] National Space Institute, Technical University of Denmark, Elektrovej, Building 328, 2800 Lyngby, Denmark. [2] Racah Institute of Physics, Hebrew University of Jerusalem, Jerusalem 91904, Israel. [3] Dark Cosmology Centre, Niels Bohr Institute, University of Copenhagen, Juliane Maries Vej 30, 2100 Copenhagen, Denmark. Correspondence and requests for materials should be addressed to H.S. (email: hsv@space.dtu.dk)

Clouds are a fundamental part of the terrestrial energy budget, and any process that can cause systematic changes in cloud micro-physics is of general interest. To form a cloud droplet, water vapor needs to condense to aerosols acting as cloud condensation nuclei (CCN) of sizes of at least 50–100 nm[1], and changes in the number of CCN will influence the cloud microphysics[2, 3]. One process that has been pursued is driven by ionization caused by cosmic rays, which has been suggested to be of importance by influencing the density of CCN in the atmosphere and thereby Earth's cloud cover[4–7]. Support for this idea came from experiments, which demonstrated that ions significantly amplify the nucleation rate of small aerosols (≈1.7 nm)[8, 9]. However, to affect cloud properties, any change in small aerosols needs to propagate to CCN sizes 50–100 nm, but such changes were subsequently found by numerical modeling to be too small to affect clouds[3, 10, 11]. The proposed explanation for this deficit is that additional aerosols reduce the concentration of the gases from which the particles grow, and a slower growth increases the probability of smaller aerosols being lost to pre-existing aerosols. This has lead to the conclusion that no significant link between cosmic rays and clouds exists in Earth's atmosphere.

This conclusion stands in stark contrast to a recent experiment demonstrating that when excess ions are present in the experimental volume, all extra nucleated aerosols can grow to CCN sizes[12]. But without excess ions in the experimental volume, any extra small aerosols (3 nm) are lost before reaching CCN sizes, in accordance with the above mentioned model results. The conjecture was that an unknown mechanism is operating, whereby ions facilitate the growth and formation of CCN. Additional evidence comes from atmospheric observations of sudden decreases in cosmic rays during solar eruptions in which a subsequent response is observed in aerosols and clouds[6, 7]. Again, this is in agreement with a mechanism by which a change in ionization translates into a change in CCN number density. However, the nature of this micro-physical link has been elusive.

In this work we demonstrate, theoretically and experimentally, the presence of an ion mechanism, relevant under atmospheric conditions, where variations in the ion density enhance the growth rate from condensation nuclei (≈1.7 nm) to CCN. It is found that an increase in ionization results in a faster aerosol growth, which lowers the probability for the growing aerosol to be lost to existing particles, and more aerosols can survive to CCN sizes. It is argued that the mechanism is significant under present atmospheric conditions and even more so during prehistoric elevated ionization caused by a nearby supernova. The mechanism could therefore be a natural explanation for the observed correlations between past climate variations and cosmic rays, modulated by either solar activity[13–17] or caused by supernova activity in the solar neighborhood on very long time scales where the mechanism will be of profound importance[18–20].

## Results

**Theoretical model and predictions.** Cosmic rays are the main producers of ions in Earth's lower atmosphere[21]. These ions interact with the existing aerosols, and charge a fraction of them. However, this fraction of charged aerosols is independent of the ionization rate in steady state—even though the electrostatic interactions enhance the interactions among the charged aerosols and between these aerosols and neutral molecules, the increased recombination ensures that the equilibrium aerosol charged fraction remains the same[22]. Ion-induced nucleation will cause the small nucleated aerosols to be more frequently charged relative to an equilibrium charge distribution, but ion recombination will move the distribution towards charge equilibrium,

typically before the aerosols reach ~4 nm[23]. Changing the ionization is therefore not expected to have an influence on the number of CCN through Coulomb interactions between aerosols.

However, this argument disregards that the frequency of interactions between ions and aerosols is a function of the ion density, and that each time an ion condenses onto an aerosol, a small mass ($m_{ion}$) is added to the aerosol. As a result, a change in ion density has a small but important effect on the aerosol growth rate, since the mass flux from the ions to the aerosols increases with the ion density. This mass flux is normally neglected when compared to the mass flux of neutral molecules (for example sulfuric acid, SA) to the aerosols by condensation growth, as can be seen from the following simple estimate: the typical ion concentration in the atmosphere is on the order of ≈$10^3$ ions cm$^{-3}$, however, the condensing vapor concentration (SA) is typically on the order of ≈$10^6$ molecules cm$^{-3}$. The ratio between them is $10^{-3}$, from which one might conclude that the effect of ions on the aerosol growth is negligible. Why this is not always the case will now be shown.

The mass flux to neutral aerosols consists not only of the condensation of neutral molecules, but also of two terms which add mass due to recombination of a positive (negative) ion and a negative (positive) aerosol. Furthermore, as an ion charges a neutral aerosol, the ion adds $m_{ion}$ to its mass. Explicitly, taking the above mentioned flux of ion mass into account, the growth of aerosols by condensation of a neutral gas and singly charged ions becomes,

$$\frac{\partial N^i(r,t)}{\partial t} = -\sum_j \frac{\partial}{\partial r} I_{i,j}(r,t) N^j(r,t),$$

$$I_{i,j}(r,t) = \begin{pmatrix} A^0 n^0 \beta^{00} & A^- n^- \beta^{-+} & A^+ n^+ \beta^{+-} \\ A^+ n^+ \beta^{+0} & A^0 n^0 \beta^{0+} & 0 \\ A^- n^- \beta^{-0} & 0 & A^0 n^0 \beta^{0-} \end{pmatrix}, \quad (1)$$

with $i$ and $j = (0, +, -)$ referring to neutral, positively, and negatively charged particles. Here $r$ and $t$ are the radius of the aerosol and the time. $N^i = (N^0, N^+, N^-)$ is the number density of neutral, positive, and negative aerosols. $n^0$ is the concentration of condensible gas, $n^+$, $n^-$ are the concentration of positive and negative ions, while $A^i = (m^i/4\pi r^2 \rho)$, with $m^i$ being the mass of the neutral gas molecule ($i = 0$), and the average mass of positive/negative ions, $i = (+, -)$, $\rho$ is the mass density of condensed gas, and $\beta$ is the interaction coefficient between the molecules (or ions) and neutral and/or charged aerosols (See Methods for details on derivation of the equations, the interaction coefficients, details of the experiment, and the ($m_{ion}/m_0$) of 2.25).

$\beta^{00}$, $\beta^{+0}$, and $\beta^{-0}$ correspond to the interaction coefficients describing the interaction between neutral aerosols of radius $r$ and neutral molecules, positive ions and negative ions respectively, whereas $\beta^{0+}$, and $\beta^{0-}$ are the interaction coefficients between neutral molecules and positively/negatively charged aerosols. Finally $\beta^{+-}$ corresponds to the recombination between a positive ion and a negative aerosol of radius $r$, and vice versa for $\beta^{-+}$[24]. If no ions are present, the above equations simplify to the well known condensation equation[25], where

$$I_{0,0}(r,t) = \frac{dr}{dt} = A^0 n^0 \beta^{00}, \quad (2)$$

is the growth rate of the aerosol radius due to the condensation of molecules onto the aerosols. It is the change in growth rate caused by ions that is of interest here.

By assuming a steady state for the interactions between ions and aerosols, we find[22]

$$\frac{N^+}{N^0} = \frac{n^+ \beta^{+0}}{n^- \beta^{-+}}, \quad \frac{N^-}{N^0} = \frac{n^- \beta^{-0}}{n^+ \beta^{+-}}, \quad (3)$$

which using $N^{\mathrm{tot}} = N^0 + N^+ + N^-$ gives

$$\frac{N^0(r,t)}{N^{\mathrm{tot}}(r,t)} = \left[ 1 + \frac{n^+ \beta^{+0}}{n^- \beta^{-+}} + \frac{n^- \beta^{-0}}{n^+ \beta^{+-}} \right]^{-1}. \quad (4)$$

Equations (3) and (4) can be inserted into the components of Eq. (1) (for $i = (0, +, -)$). Assuming symmetry between the positive and negative charges, i.e., $m_{\mathrm{ion}} \equiv m^+ = m^-$, $\beta^{\pm 0} \equiv \beta^{-0} = \beta^{+0}$, $\beta^{\pm \mp} \equiv \beta^{+-} = \beta^{-+}$, and $n_{\mathrm{ion}} \equiv n^+ = n^-$, finally leads to (See Methods for details on derivation of the equations, the interaction coefficients, details of the experiment, and the $(m_{\mathrm{ion}}/m_0)$ of 2.25):

$$\frac{\partial N^{\mathrm{tot}}(r,t)}{\partial t} = -\frac{\partial}{\partial r} \left[ A_0 n^0 \beta^{00} (1 + \Gamma) N^{\mathrm{tot}}(r,t) \right], \quad (5)$$

where

$$\Gamma = 4 \left( \frac{n_{\mathrm{ion}}}{n_0} \right) \left( \frac{\beta^{\pm 0}}{\beta^{00}} \right) \left( \frac{m_{\mathrm{ion}}}{m_0} \right) \left( \frac{N^0(r,t)}{N^{\mathrm{tot}}(r,t)} \right). \quad (6)$$

The 1 term appearing in Eq. (5) is the result of the approximation $(1 + 2(\beta^{0\pm}\beta^{\pm 0})/(\beta^{\pm \mp}\beta^{00}))/(1 + 2\beta^{\pm 0}/\beta^{\pm \mp}) \approx 1$, good to $3 \times 10^{-4}$ for a 10 nm aerosol and decreasing for $d > 10$ nm. The bracketed term in Eq. (5) is related to the rate of change in the aerosol radius

$$\frac{\mathrm{d}r}{\mathrm{d}t} = A_0 n^0 \beta^{00} [1 + \Gamma]. \quad (7)$$

This growth rate is one of the characteristic equations describing aerosol evolution, and it is valid independent of any losses[26].

It is $\Gamma$, in Eq. (6), which quantifies the net effect of ion condensation. The term $4(\beta^{\pm 0}/\beta^{00})(N^0/N^{\mathrm{tot}})$ depends on electrostatic interactions, and where $(n_{\mathrm{ion}}/n_0)$ and $(m_{\mathrm{ion}}/m_0)$ depend on the specific concentrations and parameters. Figure 1a portrays this part together with $(\beta^{\pm 0}/\beta^{00})$ and $(N^0/N^{\mathrm{tot}})$. Figure 1b depicts the size of $\Gamma$ in % of the neutral condensation, as a function of the ionization rate $q$ and diameter $d$ of the aerosols for an average atmospheric sulfuric acid concentration of $n^0 \approx 1 \times 10^6$ molecules cm$^{-3}$ and $m_0 = 100$ AMU and a mass ratio $(m_{\mathrm{ion}}/m_0)$ of 2.25 (See Methods for details on derivation of the equations, the interaction coefficients, details of the experiment, and the $(m_{\mathrm{ion}}/m_0)$ of 2.25.). It should be noted that the terms $\beta^{\pm 0}$ and $\beta^{00}$ also depend on the mass and diameter of the ions and neutral molecules, which may vary depending on composition. Both exact masses and the mass asymmetry between ions can vary—observationally positive ions tend to be heavier than negative ions[27]. There are additional caveats to the theory, which will be examined in Discussion section.

**Experimental results**. We now proceed to show that the predictions of the theory of ion-induced condensation outlined above can be measured in experiments. The latter were done in an 8 m³ stainless steel reaction chamber[12]. Due to wall losses, the growth rate of the aerosols could not be too slow, therefore the sulfuric acid concentration needed to be larger than $n^0 \approx 2 \times 10^7$ molecules cm$^{-3}$. This decreases the effect that ionization has on the aerosol growth by more than an order of magnitude when compared to typical atmospheric values. It is however a necessary constraint given the finite size of the chamber. The number of nucleated particles had to be low enough that coagulation was

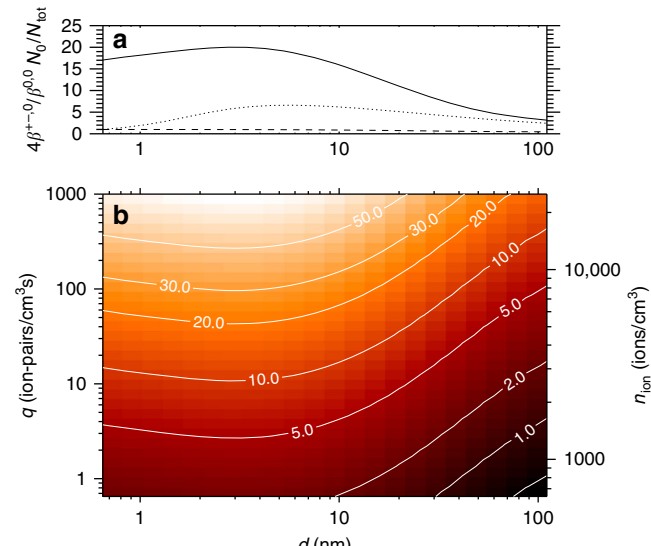

**Fig. 1** Calculation of ion contribution to growth. **a** The average relative electrostatic enhancement $4(\beta^{\pm 0}/\beta^{00})(N^0/N^{\mathrm{tot}})$ between ions and aerosols of diameter $d$ (solid line). The dotted line is $(\beta^{\pm 0}/\beta^{00})$, and the dashed line is $(N^0/N^{\mathrm{tot}})$. **b** The relative size of the ion condensation, $\Gamma$ (Eq. (6)) in %, in an atmosphere with a condensible gas concentration of $1 \times 10^6$ molecules cm$^{-3}$ as a function of aerosol diameter $d$ and ionization rate $q$ (left hand axis) or ion density (right hand axis). The contour lines show the relative size of the growth due to ion condensation in % of the usual condensation growth. The mass ratio $(m_{\mathrm{ion}}/m_0)$ is set to 2.25, and the mass of the neutral molecule is set to 100 AMU

unimportant, thus keeping the growth fronts in size-space relatively sharp, allowing accurate growth rate measurements.

The ionization in the chamber could be varied from 16 to 212 ion pairs cm$^{-3}$ s$^{-1}$ using two γ-sources. At maximum ionization, the nucleation rate of aerosols was increased by ~30% over the minimum ionization.

The experiments were performed with a constant UV photolytic production of sulfuric acid, and every 4 h (in some cases 2) the ionization was changed from one extreme to the next, giving a cycle period $P$ of 8 h (or 4) (See Methods for details on derivation of the equations, the interaction coefficients, details of the experiment, and the $(m_{\mathrm{ion}}/m_0)$ of 2.25.). The effect of ion-induced nucleation during the part of the cycle with maximum ionization results in an increased formation of new aerosols (Fig. 2a). To improve the statistics, the cycle $P$ was repeated up to 99 times. A total of 11 experimental runs were performed, representing 3100 h. Each data set was subsequently superposed over the period $P$ resulting in a statistically averaged cycle. An example of a superposed cycle can be seen in Fig. 2b), where locations of the transition regions between the low and high aerosol density data can be used to extract the effect of ions on aerosols growth. The two transitions determine two trajectories, profile 1 and profile 2, in the $(d, t)$-plane, from which it is possible to estimate the difference in the growth time to a particular size $d$ (See Methods for details on derivation of the equations, the interaction coefficients, details of the experiment, and the $(m_{\mathrm{ion}}/m_0)$ of 2.25.). A CI API-ToF mass spectrometer was used to measure the sulfuric acid concentration during some of the experiments and to estimate the average ion mass[28].

The above theory predicts a difference in the time it takes the two profiles to reach a size $r$ due to a growth velocity difference caused by ion condensation. The time it takes for aerosols to grow

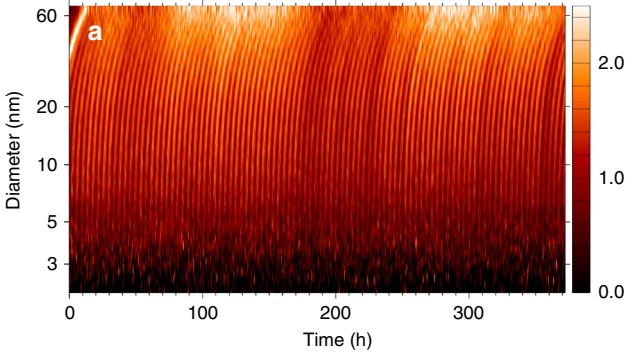

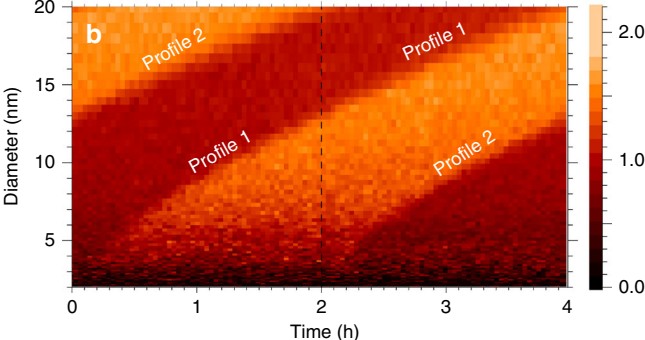

**Fig. 2** Experimental growth profiles. **a** Aerosol number density $N^{\text{tot}}(d,t)/\left(1/T\int_0^T N^{\text{tot}}(d,t')dt'\right)$, normalized by the average of 97 cycles of 4 h ($T = 388$ h), as a function of diameter $d$ and time $t$. **b** Data superposed over the 4 h period. The time $t_1(d)$ (or $t_2(d)$) that the profile 1 (or profile 2) reaches $d$ is determined by the local maximum of a Gaussian fit to ($dN^{\text{tot}}(d, t)/dt)^2$ (See Methods for details on derivation of the equations, the interaction coefficients, details of the experiment, and the ($m_{\text{ion}}/m_0$) of 2.25.). Note that profile 1 (profile 2) is initially growing with γ-on (γ-off) until $d \approx 13$ nm. However when $d > 13$ nm profile 1 (profile 2) grows with γ-off (γ-on). It is the difference in timing of profile 1 and 2 that contain information about the effect of ions on the growth rate

to size $r$ along the two possible profiles is expressed as

$$t_i(r) = \int_0^r \left[\left(\frac{dr}{dt}\right)_i\right]^{-1} dr, \qquad (8)$$

where $t_1$ and $t_2$ refers to the time it takes profiles 1 and 2 to reach size $r$. The integrand is given by Eq. (7) and it considers that after half the period, the γ-sources are switched off (or on). The above equations can be integrated numerically to find $\Delta T = t_2(r) - t_1(r)$ and allow comparison with the experiments.

During the first ~12 nm of growth, profile 1 grows with the γ-sources on and it thus grows faster than profile 2 in the γ-off region, consequently, $t_1(r) < t_2(r)$ and $\Delta T$ is increasing (Fig. 2b). This increase is due to the (nearly) constant difference in growth rate between the two profiles. But when profile 1 enters the second part of the cycle, when the γ-sources are off, profile 2 enters the high ion state and is now growing faster than profile 1. Therefore, it is now profile 2 that grows faster and $\Delta T$ starts to decrease. Figure 3 depicts three examples of $\Delta T$ as a function of the diameter $d$. It is seen that the data scatter around the theoretical curves (red (γ-on) and blue (γ-off)) obtained from Eqs. (7) and (8). The gray curves were produced by performing a LOESS (locally weighted smoothing) smoothing of the experimental data. It also indicates that the enhanced growth is continuing up to at least 20 nm, and in good agreement with theory. Note that although some of the experiments contain size

distribution data above 20 nm, the profiles at those sizes become poorly defined at which point we stop the analysis.

All 11 experimental runs are summarized in Fig. 4, where $\Delta T$ is averaged between 6 and 12 nm, and shown as a function of the SA concentration, which is obtained from either CI-API-ToF measurements and/or slopes of the growth profiles. The red curve is the theoretical expectation for the γ-sources at maximum, and the blue curve is obtained with a 45% reduction in the ion density. Both are found by numerically solving Eqs. (7) and (8). The relative importance of ion condensation increases as the SA concentration is lowered, as predicted and in good agreement with theory.

## Discussion

The most common effect of ions considered in aerosol models is aerosol charging which increases the interaction between the charged aerosols and neutral aerosols/molecules, thereby increasing aerosol growth. However, as mentioned previously, the ion density does not affect the steady state fraction of aerosols that are charged such that the ion-induced interactions remain nearly constant, implying that no effect on the aerosol growth is expected by changing the background ionization. Nonetheless, experiments and observations do suggest that ions have an effect on the formation of CCN, the question has therefore been, how is this possible?

The present work demonstrates that the mass flux associated with the aerosol charging by ions and ion–aerosol recombination is important and should not be neglected. $\Gamma$ in Eq. (7) contains the effect of the mass-flux of ions to aerosols and demonstrates the inherent amplifications by the interaction between the ions and aerosols. This function $\Gamma$ shows that the initial estimate of the mass-flux, $(n_{\text{ion}}/n^0) = 10^{-3}$, made in the introduction, gets multiplied by the size-dependent function $4\left(\beta^{\pm 0}/\beta^{00}\right)\left(\frac{m_{\text{ion}}}{m_0}\right)(N_0/N_{\text{tot}})$ which at maximum is about 60 ($m_{\text{ion}}/m_0 \sim 2.25$), and therefore nearly two orders of magnitude larger, than the naive estimate. The simple expression for the growth rate, Eq. (7), can conveniently be used as a parametrization in global aerosol models.

As a test of the theoretical model, extensive experiments were performed to study the effect on growth of the flux of ion-mass to the aerosols. One complication in the experiments was that aerosols were lost to the walls of the chamber. This meant that the concentration of SA could not be as low as the typical values in the atmosphere ~$10^6$ molecules cm$^{-3}$, but had to be higher than ~$2 \times 10^7$ molecules cm$^{-3}$. Therefore, the relative effect on the growth caused by the ions was more than an order of magnitude smaller, as can be seen from Eq. (7). The experimental challenge was therefore to measure a <1% change in growth rate, which was done by cyclic repeating the experiments up to 99 times and average the results in order to minimize the fluctuations, with a total of 3100 h of experiments. Figures 3 and 4 demonstrate both the importance of varying the neutral SA gas concentration and the effect of changing the ion density, and show excellent agreement with the theoretical expectations. One important feature is that the effect on the growth rate continues up to ~20 nm, as can be seen in Fig. 3, which is larger sizes than predicted for charged aerosols interacting with neutral molecules[29-31], and is expected to increase for atmospherically relevant concentrations of SA. It should be noted that the early stages of growth are very important since the smallest aerosols are the most vulnerable to scavenging by large pre-existing aerosols, and by reaching larger sizes ~20 nm faster, the survivability increases fast.

The presented theory is an approximation to a complex problem, and a number of simplifications have been made which gives rise to some questions. We will now discuss the most

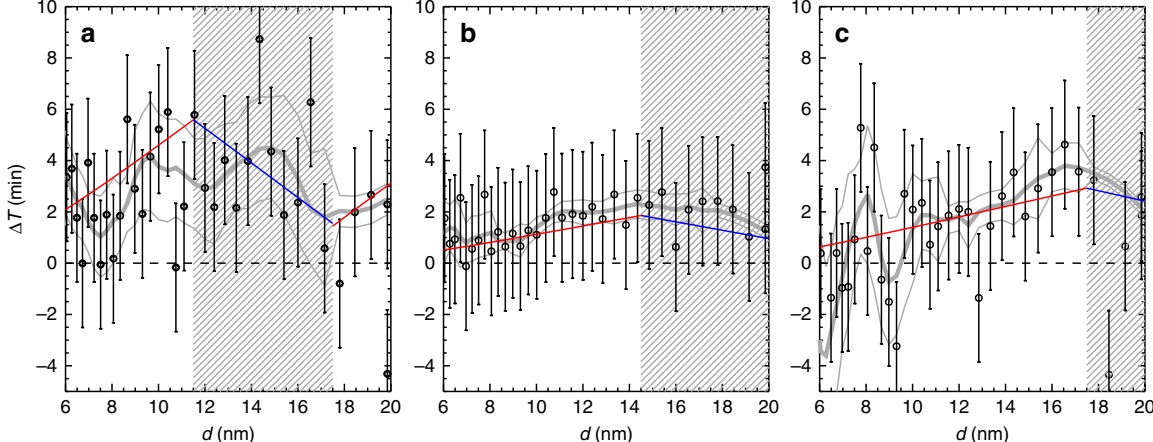

**Fig. 3** Temporal evolution of growth time difference. Three examples of growth time differences $\Delta T$ versus diameter is shown as black diamond symbols along with 1 std. dev. uncertainty. Red (Blue) curves are the theoretical expectations during gamma on(off) periods based on numerical integration of Eq. (7). **a** Experimental run V9 (Fig. 4), based on 45 cycles of 8 h. **b** Experimental run V11, based on 99 cycles of 8 h. **c** Experimental run V7, based on 4 cycles of 8 h. The hatched regions denotes growth periods in the γ-off state. The gray curves are a LOEES smoothing of the experimental data, together with the 1 std. dev. uncertainty. The scattering of points is smallest for run V11, which has the most cycles

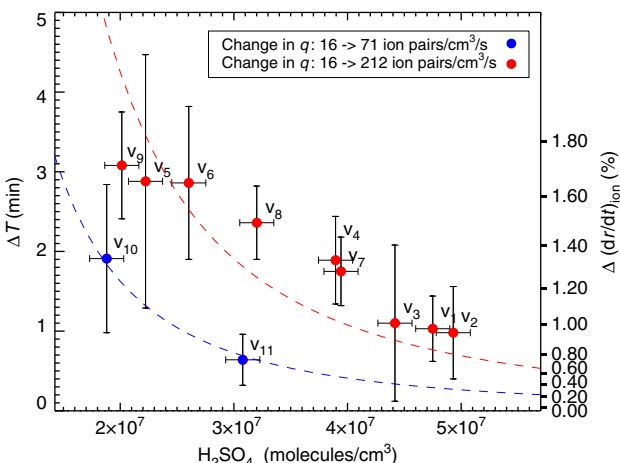

**Fig. 4** Comparison of experiments and theory. Time difference between γ-on profile and γ-off profile averaged between $d = 6$ nm and $d = 12$ nm as a function of sulfuric acid concentration. The red circle symbols are for $\Delta q = 196$ ion pairs cm$^{-3}$ s$^{-1}$ and blue circles are for $\Delta q = 55$ ion pairs cm$^{-3}$ s$^{-1}$. Error bars are the 1 std. dev. uncertainty. The red (blue) curve is the theoretical expectation based on Eqs. (7) and (8). Right-hand axis is the relative change in growth rate averaged between $d = 6$ nm and $d = 12$ nm, in %

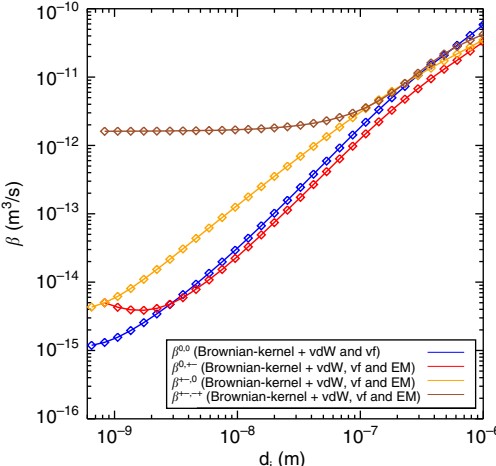

**Fig. 5** Interaction coefficients. The interaction coefficients between a small neutral particle of mass 100 AMU and a small ion of mass 225 AMU interacting with aerosols of diameter $d$. The interaction between neutral particles, $\beta^{00}$, is given by the blue curve, the interaction between small neutral particles and charged aerosols, $\beta^{0\pm}$, is given by the red curve. The interaction between a positive or negative ion and neutral aerosols, $\beta^{\pm 0}$, is described with the yellow curve. Finally, the recombination coefficient between two oppositely charged particles is given by the brown curve. The coefficients were calculated assuming Brownian diffusion while including Van der Waals-forces, Coulomb-forces (including image charges) and viscous forces[24]. Symmetry between positive and negative ions has been assumed, see text

pertinent: Will the material that constitute the ions condense onto the aerosols in any case as neutral molecules? This will certainly be the case for the negative HSO$_4^-$ ions. Assuming that all negative ions, $n^-$, are HSO$_4^-$, then the number of neutral SA molecules would be $n_0 - n^-$, where $n^-$ is the total negative ion density. Inserting values in the right hand side of Eq. (7), for example for the present experiment $n_0 \sim 10^7$ molecules cm$^{-3}$, and $n^- \sim 10^4$ ions cm$^{-3}$ the correction to the growth rate from the decrease in neutral molecules is, $|\Delta(\mathrm{d}r/\mathrm{d}t)/(\mathrm{d}r/\mathrm{d}t)| \sim |((n^0 - n^-) - n^0)/n^0| < 10^{-3}$, but the ion condensation impact on the growth rate is of the order $10^{-2}$ (Fig. 4) and therefore an order of magnitude smaller. So even if the neutral molecules would condense eventually, it does not change the estimated growth rate by ion condensation significantly. This would also be the case under atmospheric conditions, where $n_0$ is of the order

$10^6$ cm$^{-3}$ and $n_{ion} \sim 10^3$ ions cm$^{-3}$, again a correction an order of magnitude lower than the ion condensation effect. Also note that the mass-flux from ions is larger than from the neutral molecules, which is part of the faster growth rate. In fact, even if the larger particles grow slightly slower due to a decrease in neutral molecules, the growth rate of the smaller particles is enhanced due to the ion interactions, which make the cross-section of the small particles larger (Fig. 5). This leads to the second question: Will the ion-mass that condenses onto the small aerosols stay in the aerosol and not evaporate after the aerosol is neutralized? This is slightly more difficult to answer, since the composition of all the

ions are not known. The abundant terminal negative $HSO_4^-$ ions are not more likely to evaporate than the neutral SA molecules. With respect to unknown positive or negative ions the possibility of evaporation is more uncertain. If the material of some of the ions are prone to evaporate more readily, it would of course diminish the ion effect. The present experimental conditions did not indicate that this was a serious problem, but in an atmosphere of e.g. more volatile organics it could be. Another issue is that sulfate ions typically carry more water than their neutral counterparts[32], and it is uncertain what happens with this excess water after neutralization of the aerosol. It was also assumed that the ion density was in steady state with the aerosol density at all times. This is of course an approximation, but from measurements of the ion density with a Gerdien tube[33] the typical time scale for reaching steady state is minutes and the assumption of an ion density in steady state is thus a reasonable approximation[12]. It is worth noting that in the experiments two types of losses for ions are present, in addition to recombination: Wall losses and condensation sink to aerosols. Based on the loss rate of sulfuric acid the wall loss rate is about $7 \times 10^{-4}\,s^{-1}$, while the condensation sink for experiment V2 was $1.2 \times 10^{-4}\,s^{-1}$. This means that the wall losses were dominant and changes in the aerosol population will thus have a minimal influence on the ion concentration. Furthermore recombination is by far the dominant loss mechanism for ions. For an ion production rate of $16\,cm^{-3}\,s^{-1}$, the actual ion concentration is 92% of what a calculation based only on recombination gives—for larger ion production the recombination becomes more dominant and vice versa. Under atmospheric conditions of high condensation sink and low ion production this may constitute a significant decrease to the effect due to the reduced ion concentration, but under clean conditions and in the experiment the condensation sink has an minor effect. In order to calculate the interaction coefficients between ions and aerosols it is necessary to know the mass of the ions and mass of the aerosols. This is complex due to the many ion species and their water content, and as a simplification an average ion mass was chosen to be 225 AMU. The sensitivity of the theory to changes in ion mass in the range (130–300 AMU) and mass of a neutral SA molecule in the range (100–130) could change the important ratio ($\beta^{\pm 0}/\beta^{00}$) by up to 20%.

The possible relevance of the presented theory in Earth's atmosphere will now be discussed. From Eq. (6), the factor ($n_{ion}/n^0$) indicates that the relative importance of ion condensation will be largest when the concentration of condensing gas $n_0$ is small and the ion density is large. Secondly, the number density of aerosols should also be small so the majority of ions are not located on aerosols. This points to pristine marine settings over the oceans, away from continental and polluted areas. Results based on airborne measurements suggest that the free troposphere is a major source of CCN for the Pacific boundary layer, where nucleation of new aerosols in clean cloud processed air in the Inter-Tropical Convergence Zone are carried aloft with the Hadley circulation and via long tele-connections distributed over $\sim \pm\,30°$ latitude[34, 35]. In these flight measurements, the typical growth rate of aerosols was estimated to be of the order $\sim0.4\,nm\,h^{-1}$[35], which implies an average low gas concentration of condensing gas of $n_0 \sim 4 \times 10^6$ molecules $cm^{-3}$. Measurements and simulations of SA concentration in the free troposphere annually averaged over day and night is of the order $n_0 \sim 10^6$ molecules $cm^{-3}$[36]. This may well be consistent with the above slightly larger estimate, since the aerosol cross-section for scavenging smaller aerosols increases with size, which adds to the growth rate. Secondly, the observations suggest that as the aerosols enters the marine boundary layer, some of the aerosols are further grown to CCN sizes[35]. Since the effect of ion condensation scales inversely with $n_0$, a concentration of $n_0 \sim 4 \times 10^6$ molecules $cm^{-3}$ would

diminish the effect by a factor of four. As can be seen in Fig. 1b, the effect of ion condensation for an ionization rate of $q = 10$ ion pairs $cm^{-3}\,s^{-1}$ would change from 10 to 2.5% which may still be important. Note that other gases than sulfuric acid can contribute to $n_0$ in the atmosphere. As aerosols are transported in the Hadley circulation, they are moved in to the higher part of the troposphere, where the intensity and variation in cosmic rays ionization are the largest[37]. This suggests that there are vast regions where conditions are such that the proposed mechanism could be important, i.e., where aerosols are nucleated in Inter-Tropical Convergence Zone and moved to regions where relative large variations ionization can be found. Here the aerosols could grow faster under the influence of ion condensation, and the perturbed growth rate will influence the survivability of the aerosols and thereby the resulting CCN density. Finally the aerosols are brought down and entrained into the marine boundary layer, where clouds properties are sensitive to the CCN density[2].

Although the above is on its own speculative, there are observations to further support the idea. On rare occasions the Sun ejects solar plasma (coronal mass ejections) that may pass Earth, with the effect that the cosmic ray flux decreases suddenly and stays low for a week or two. Such events, with a significant reduction in the cosmic rays flux, are called Forbush decreases, and can be used to test the link between cosmic ray ionization and clouds. A recent comprehensive study identified the strongest Forbush decreases, ranked them according to strength, and discussed some of the controversies that have surrounded this subject[7]. Atmospheric data consisted of three independent cloud satellite data sets and one data set for aerosols. A clear response to the five strongest Forbush decreases was seen in both aerosols and all low cloud data[7]. The global average response time from the change in ionization to the change in clouds was $\sim7$ days[7], consistent with the above growth rate of $\sim0.4\,nm\,h^{-1}$. The five strongest Forbush decreases (with ionization changes comparable to those observed over a solar cycle) exhibited inferred aerosol changes and cloud micro-physics changes of the order $\sim2\%$[7]. The range of ion production in the atmosphere varies between 2 and 35 ions pairs $s^{-1}\,cm^{-3}$[37] and from Fig. 1b it can be inferred from that a 20% variation in the ion production can impact the growth rate in the range 1–4% (under the pristine conditions). It is suggested that such changes in the growth rate can explain the $\sim2\%$ changes in clouds and aerosol change observed during Forbush decreases[7]. It should be stressed that there is not just one effect of CCN on clouds, but that the impact will depend on regional differences and cloud types. In regions with a relative high number of CCN the presented effect will be small, in addition the effect on convective clouds and on ice clouds is expected to be negligible. Additional CCNs can even result in fewer clouds[38]. Since the ion condensation effect is largest for low SA concentrations and aerosol densities, the impact is believed to be largest in marine stratus clouds.

On astronomical timescales, as the solar system moves through spiral-arms and inter-arm regions of the Galaxy, changes in the cosmic ray flux can be much larger[18–20]. Inter-arm regions can have half the present day cosmic ray flux, whereas spiral arm regions should have at least 1.5 times the present day flux. This should correspond to a $\sim10\%$ change in aerosol growth rate, between arm and inter-arm regions. Finally, if a near-Earth supernova occurs, as may have happened between 2 and 3 million years ago[39], the ionization can increase 100 to 1000 fold depending on its distance to Earth and time since event. Figure 1b shows that the aerosol growth rate in this case increases by more than 50%. Such large changes should have profound impact on CCN concentrations, the formation of clouds and ultimately climate.

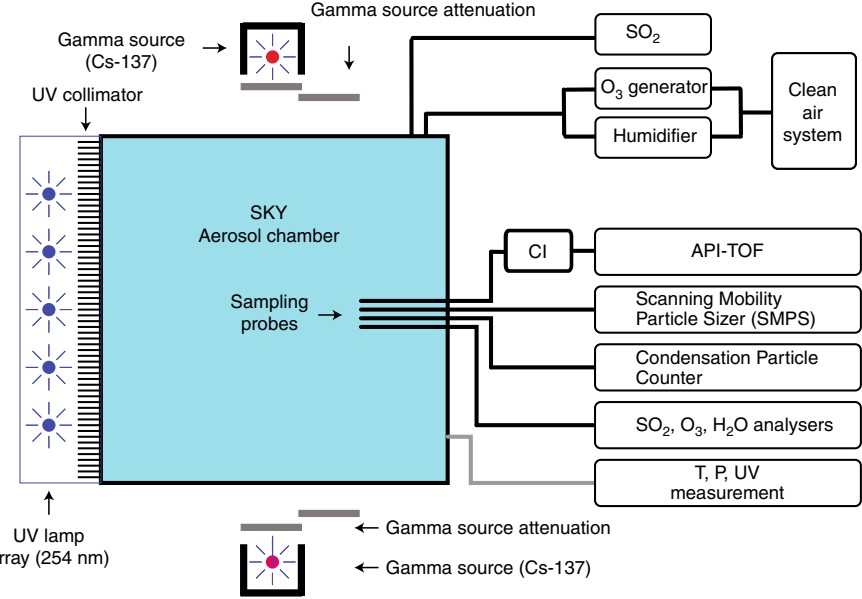

**Fig. 6** The experimental setup

In conclusion, a mechanism by which ions condense their mass onto small aerosols and thereby increase the growth rate of the aerosols, has been formulated theoretically and shown to be in good agreement with extensive experiments. The mechanism of ion-induced condensation may be relevant in the Earth's atmosphere under pristine conditions, and able to influence the formation of CCN. It is conjectured that this mechanism could be the explanation for the observed correlations between past climate variations and cosmic rays, modulated by either solar activity[13–17] or supernova activity in the solar neighborhood on very long time scales[18–20]. The theory of ion-induced condensation should be incorporated into global aerosol models, to fully test the atmospheric implications.

## Methods

**Correction to condensation due to ions.** Expanding Eq. (1) gives

$$\frac{\partial N^0}{\partial t} = -\frac{\partial}{\partial r}\left(A^0 n^0 \beta^{00} N^0 + A^- n^- \beta^{-+} N^+ + A^+ n^+ \beta^{+-} N^-\right)$$
$$\frac{\partial N^+}{\partial t} = -\frac{\partial}{\partial r}\left(A^0 n^0 \beta^{0+} N^+ + A^+ n^+ \beta^{+0} N^0\right) \qquad (9)$$
$$\frac{\partial N^-}{\partial t} = -\frac{\partial}{\partial r}\left(A^0 n^0 \beta^{0-} N^- + A^- n^- \beta^{-0} N^0\right).$$

where the indexes 0, +, and − refer to neutral, positively, and negatively charged particles. Here $r$ and $t$ are the radius of the aerosol and the time. $N^0$, $N^+$, and $N^-$ the number density of neutral, positive, and negative aerosols. $n^0$ is the concentration of the condensible gas (usually sulfuric acid in the gas phase), $n^+$ and $n^-$ are the concentration of positive and negative ions, $A^0 = (m^0/4\pi r^2\rho)$, $A^+ = (m^+/4\pi r^2\rho)$, and $A^- = (m^-/4\pi r^2\rho)$, where $m^0$ is the mass of the neutral gas molecule, $m^+$ and $m^-$ are the average mass of positive/negative ions, $\rho$ is the mass density of condensing gas, and $\beta$ the interaction coefficient between the monomers and the neutral and/or charged aerosols. The parameters of the above model are shown in Fig. 5.

Using equilibrium between aerosols and ions we have

$$\frac{N^+}{N^0} \approx \frac{n^+ \beta^{+0}}{n^- \beta^{-+}}, \qquad \frac{N^-}{N^0} \approx \frac{n^- \beta^{-0}}{n^+ \beta^{+-}}, \qquad (10)$$

while defining $N^{\text{tot}} = N^0 + N^+ + N^-$ gives

$$\frac{N^0(r,t)}{N^{\text{tot}}(r,t)} = \left[1 + \frac{n^+ \beta^{+0}}{n^- \beta^{-+}} + \frac{n^- \beta^{-0}}{n^+ \beta^{+-}}\right]^{-1}. \qquad (11)$$

If we further assume symmetry between the positive and negative charges, i.e., that $m_{\text{ion}} \equiv m^+ = m^-$, $\beta^{\pm 0} \equiv \beta^{-0} = \beta^{+0}$, $\beta^{\pm\mp} \equiv \beta^{+-} = \beta^{-+}$ as well as $n_{\text{ion}} \equiv n^+ = n^-$, such that $A^\pm \equiv A^+ = A^-$, we find

$$\frac{N^\pm}{N^0} = \frac{\beta^{\pm 0}}{\beta^{\mp \pm}}, \qquad (12)$$

and for $N^{\text{tot}} = N^0 + N^+ + N^-$, we obtain

$$\frac{N^0(r,t)}{N^{\text{tot}}(r,t)} = \left[1 + 2\frac{\beta^{\pm 0}}{\beta^{\mp \pm}}\right]^{-1}. \qquad (13)$$

Using Eq. (12) in Eq. (9) and using the charge symmetry gives

$$\frac{\partial N^0}{\partial t} = -\frac{\partial}{\partial r}\left(\left[A^0 n^0 \beta^{00} + 2A^\pm n_{\text{ion}} \beta^{\pm 0}\right] N^0\right)$$
$$\frac{\partial N^+}{\partial t} = -\frac{\partial}{\partial r}\left(\left[A^0 n^0 \beta^{0\pm} \frac{\beta^{\pm 0}}{\beta^{\mp \pm}} + A^\pm n_{\text{ion}} \beta^{\pm 0}\right] N^0\right)$$
$$\frac{\partial N^-}{\partial t} = -\frac{\partial}{\partial r}\left(\left[A^0 n^0 \beta^{0\pm} \frac{\beta^{\pm 0}}{\beta^{\mp \pm}} + A^\pm n_{\text{ion}} \beta^{\pm 0}\right] N^0\right).$$

Adding the three equations then results in

$$\frac{\partial N^{\text{tot}}}{\partial t} = -\frac{\partial}{\partial r}\left(\left[A^0 n^0\left(\beta^{00} + 2\beta^{0\pm}\frac{\beta^{\pm 0}}{\beta^{\mp \pm}}\right) + 4A^\pm n_{\text{ion}} \beta^{\pm 0}\right] N^0\right). \qquad (14)$$

Using $N^{\text{tot}}$ as a common factor, we then have

$$\frac{\partial N^{\text{tot}}}{\partial t} = -\frac{\partial}{\partial r}\left(\left[A^0 n^0\left(\beta^{00} + 2\beta^{0\pm}\frac{\beta^{\pm 0}}{\beta^{\mp \pm}}\right)\frac{N^0}{N^{\text{tot}}} + 4A^\pm n_{\text{ion}} \beta^{\pm 0}\frac{N^0}{N^{\text{tot}}}\right]N^{\text{tot}}\right). \qquad (15)$$

Taking $\beta^{00}$ as a common factor and plugging Eq. (13) into the first term gives the expression

$$F = \frac{\left(1 + 2\beta^{0\pm}\beta^{\pm 0}/(\beta^{\mp \pm}\beta^{00})\right)}{\left[1 + 2\beta^{\pm 0}/\beta^{\mp \pm}\right]}. \qquad (16)$$

The above function is equal to $1 + \mathcal{O}(10^{-2})$, and $F$ is therefore replaced with 1. A simple rearrangement provides the final form

$$\frac{\partial N^{\text{tot}}(r,t)}{\partial t} = -\frac{\partial}{\partial r}\left[A_0 n^0 \beta^{00}(1 + \Gamma)N^{\text{tot}}(r,t)\right], \qquad (17)$$

where

$$\Gamma = 4\left(\frac{n_{\text{ion}}}{n_{\text{sa}}}\right)\left(\frac{\beta^{\pm 0}}{\beta^{00}}\right)\left(\frac{m_{\text{ion}}}{m^0}\right)\left(\frac{N^0(r,t)}{N^{\text{tot}}(r,t)}\right). \qquad (18)$$

**Detailed description of the experimental setup.** The experiments were conducted in a cubic 8 m³ stainless steel reaction chamber used in Svensmark et al.[12], and shown schematically in Fig. 6. One side of the chamber is made of Teflon foil to allow the transmission of collimated UV light (253.7 nm), that was used for photolysis of ozone to generate sulfuric acid that initiates aerosol nucleation. The chamber was continuously flushed with 20 L min⁻¹ of purified air passing through a humidifier, 5 L min⁻¹ of purified air passing through an ozone generator, and 3.5 mL min⁻¹ of SO₂ (5 ppm in air, AGA). The purified air was supplied by a compressor with a drying unit and a filter with active charcoal and citric acid.

The chamber was equipped with gas analyzers for ozone and sulfur dioxide (a Teledyne 400 and Thermo 43 CTL, respectively) and sensors for temperature and

## Table 1 Overview of experimental runs

| Exp.[a] | P[b] | N[c] | Scan range[d] | UV[e] | RH[f] | CPC[g] | Lead[h] |
|---|---|---|---|---|---|---|---|
| — | h | # | nm | % | % | Model | cm |
| V1* | 4 | 23 | 3.5–118 | 80 | 14 | 3775 | 0 |
| V2* | 4 | 97 | 2–63.8 | 70 | 23 | 3776 | 0 |
| V3* | 8 | 16 | 2–63.8 | 70 | 23 | 3776 | 0 |
| V4* | 4 | 77 | 2–63.8 | 50 | 23 | 3776 | 0 |
| V5* | 8 | 44 | 2–63.8 | 40 | 15 | 3775 | 0 |
| V6 | 8 | 22 | 2–63.8 | 35 | 21 | 3775 | 0 |
| V7 | 8 | 4 | 4.0–20.2 | 35 | 37 | 3775 | 0 |
| V8 | 8 | 12 | 4.0–20.2 | 25 | 38 | 3775 | 0 |
| V9 | 8 | 45 | 4.0–20.2 | 15 | 38 | 3775 | 0 |
| V10 | 8 | 47 | 4.0–20.2 | 15 | 38 | 3775 | 1 |
| V11 | 8 | 99 | 4.0–20.2 | 25 | 37 | 3775 | 1 |

[a]Shows the name of the experiment, used for reference. An asterisk (*) next to the name indicates that sulfuric acid was measured during the experiment

[b]Length of the period ($P$) where a $P$ of 4 h means that the experiment had 2 h of γ-rays on and 2 h of γ-rays off

[c]Number of repetitions (periods) of the experiment

[d]Scan range of the DMA, which was narrowed in later runs without changing the scan-time to improve counting statistics

[e]Setting of the UV light used to produce sulfuric acid, in percentage of maximum power.

[f]Relative humidity in the chamber

[g]TSI model number of the CPC used

[h]Amount of lead in front of the gamma sources during the gamma-on time

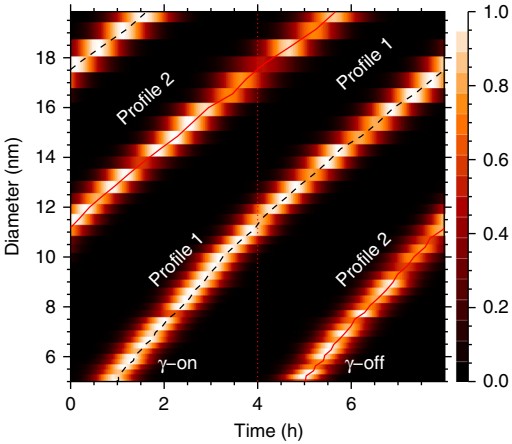

**Fig. 8** Example of $(d(N^{tot}(d,t)/\langle N(d)\rangle)/dt)^2$, normalized with this functions maximum value at diameter $d$, in the $(d,t)$-plane. From experiment V9. The black dashed line and red lines are the maximum values, found from a Gaussian fit, and determine the evolution of the profiles 1 and 2

## Table 2 Average mass spectra

| UV | Gamma | Mass | Mass w. water |
|---|---|---|---|
| % | — | $m/q$ | $m/q$ |
| 0 | Off | 258 | 280 |
| 25 | Off | 177 | 214 |
| 25 | On | 174 | 209 |
| 50 | Off | 189 | 227 |
| 70 | Off | 183 | 220 |
| 70 | On | 175 | 212 |

Each line shows the conditions and average $m/q$ for a 4-h API-ToF mass spectrum without the CI. Column 1 shows the UV level as percentage of maximum power. Column 2 shows whether the γ-ray sources were on or off. Column 3 is the average $m/q$ of the spectrum. Column 4 is the average mass of the spectrum, when 1 water ($m/q$ 18) has been added to all masses except the first four sulfuric acid peaks ($m/q$ 97, 195, 293, 391) which has 1.5 water per sulfuric acid

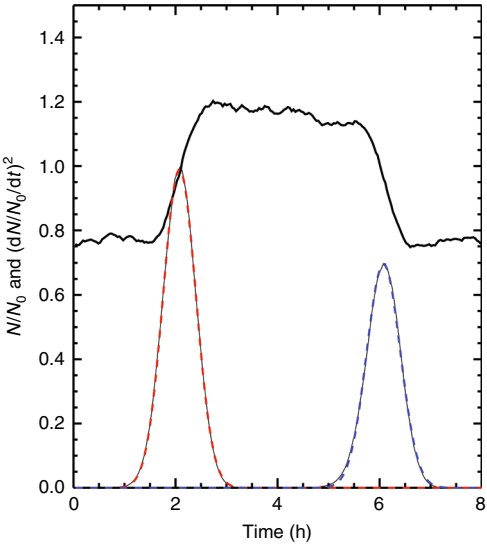

**Fig. 7** Superposed data and gaussian fits using the data from experiment V9 and SMPS size-bins centered on $d = 7.51$ nm

relative humidity. For aerosol measurements, a scanning mobility particle sizing (SMPS) system was used. The system consisted of an electrostatic classifier (TSI model 3080 with a model 3077A Kr-85 neutralizer) using a nano-DMA (TSI model 3085) along with either one of two condensation particle counters (TSI model 3775 or 3776). For some of the experiments, a CI API-ToF[28] using HNO₃ as the ionizing agent was used to measure the sulfuric acid in the chamber. The ionization in the chamber could be increased by two 27 MBq Cs-137 gamma sources placed 0.6 m from opposing sides of the chamber, with the option of putting attenuating lead plates of 0.5, 1.0, and 2.0 cm thickness in front of each source. At full strength the sources increase the ionization in the chamber to 212 ion pairs cm$^{-3}$ s$^{-1}$.

**Details of the data analysis.** A total of 11 experimental runs totaling 3100 h of measurements were made with varying settings. The settings for each of the experiments are shown in Table 1.

To detect an eventual difference in growth rate the following method was employed. For each experimental run each size-bin was normalized and then the individual periods were superposed to reduce the noise in the data, as shown in Fig. 2 of the main paper. The superposed data was then used for further analysis. For each size-bin recorded by the SMPS, the number of aerosols relative to the

mean number $\left(\langle N(d)\rangle = 1/T \int_0^T N^{tot}(d,t')dt'\right)$ was then plotted—as exemplified in the top curve of Fig. 7. The derivative of this curve, is the rate of change of aerosol density of a given size, is used to determine the temporal position of the profiles 1 and 2. This can be achieved by first calculating the derivative $\left((d(N^{tot}/\langle N(d)\rangle)/dt)^2\right)$, then normalizing with this function's maximum value at diameter $d$, (the square was used to get a positive definite and sharply defined profile), and then smoothed using a boxcar filter with a width of typically 7–16 min —shown as the lower black curve in Fig. 7. The width of the boxcar filter was typically determined from the requirement that the Gaussian fit converged—for instance, in some cases with low sulfuric acid concentration a longer boxcar filter was used, due to the relatively higher noise.

On top of the black curve in Fig. 7, a dashed red and a dashed blue curve are superimposed. These are Gaussian fits to the two maxima. The position of the center of each of the Gaussian profiles gives the growth time relative to the time the γ sources were opened (profile 1) or closed (profile 2). Figure 8 plots $(d(N^{tot}/\langle N(d)\rangle)/dt)^2$, normalized with this functions maximum value at diameter $d$, in the $(d,t)$-plane. The position of the maxima are easily seen. The black dashed and red curves in Fig. 8 are the maxima obtain from the Gaussian fits of profile 1 and profile 2.

The difference between these growth times then gives the $\Delta T$ for each bin size, as shown in Fig. 3. The $\Delta T$ values can then be compared with the theoretical expectations. Averaging the individual $\Delta T$ values for sizes between 6 and 12 nm finally results in the $\Delta T$ shown in Fig. 4.

**The $m_{ion}/m_0$ ratio.** Table 2 summarizes the average masses ($m/q$) of a series of runs using the API-ToF without the CI-unit to measure negative ions in order to determine the ratio $m_{ion}/m_0$. Note that water evaporates in the API-ToF so the masses measured are lower than the actual masses of the clusters. The ratio of 2.25 for $m_{ion}/m_0$ used in the calculations would imply that for a dry (0 water) neutral sulfuric acid molecule (98 AMU) $m_{ion}$ should be 220 $m/q$. The amount of water on a sulfuric acid molecule varies according to relative humidity—for 50% RH it is typically 1–2 water molecules. Assuming 1.5 waters and $m_{ion}/m_0 = 2.25$ this would

give a wet mass of 281 AMU. However, the experiments were performed at lower RH than 50% and also note that hydrogen sulfate ions attract more water than the neutral sulfuric acid molecule[32]. Last, the positive ions were not measured and these are typically heavier than the negative ions[27].

**Data availability**. The data generated during the current study are available from the corresponding author on reasonable request.

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

## Acknowledgements

Jacob Svensmark is funded by the Danish council for independent research under the project Fundamentals of Dark Matter Structures, DFF 6108-00470. H.S. thanks the late Nigel Calder for many discussions in the early part of this work.

## Author contributions

H.S. made the theoretical calculations, designed and helped with the experiments and wrote the first draft of the paper. M.B.E. made the experiments and made input to the paper. N.J.S. made input to the theory, experiments, and paper. J.S. helped with the experiments, calculated the interaction coefficients, and made input to the paper.

## Additional information

**Competing interests:** The authors declare no competing financial interests.

