## [Peer Review File · Nature Communications]

Reviewer #1 (Remarks to the Author):

Review by Jeff Pierce (From the journal's reviewer instructions: "To increase the transparency and openness of the reviewing process, we do support our reviewers signing their reports to authors if the reviewers feel comfortable doing so.")

This manuscript presents theory and experimental evidence that small atmospheric ions provide non-trivial condensational mass to growing ultrafine particles. The authors also provide a back-of-envelope estimate of how the theory and experiments apply to the atmosphere as a whole with estimates of how modulations in cosmic rays might affect CCN concentrations

This is a very interesting manuscript. In my opinion, the experimental results provide a bedrock for everything else, and they are the centerpiece of the work. The authors seemed to have dedicated a tremendous amount of time/effort into these experiments to bring out a clear signal in the difference in growth rates with ion concentrations. I would like these experimental results to be published to stimulate related research throughout the community.

I see several potential issues with the ion-condensation theory developed in the paper (the role of condensation sink connecting ion formation to ion concentrations; the fraction of ionic mass that would have condensed anyway if it were not in ions e.g. sulfuric acid in ions; and the irreversibility of the condensation of other species included in the ions). I discuss these issues in detail below. I think these are major issues that the authors need to address, but I think that there may be ways to update the paper to clearly state assumptions and potential limitations of the theory (e.g. to present this theory as a starting point), and to show that even with these limitations, the theory appears to describe the experiments.

I also find issues with the calculations of the global atmospheric impact of ion condensation on CCN concentrations that I describe in detail below. At least one of these issues (the failure to consider primary emissions as a source of many of the CCN in the atmosphere) would cause the estimated numbers here to be overestimated (I would guess by about a factor of 2). The other issues (the lack of a consideration [or at least an explicit discussion] of the variability in the condensation sink of ions, and the error in equation 9) may influence the calculation up or down (I am unable to determine the sign, especially for the condensation sink issue since I don't know what was assumed for this condensation sink to translate ion formation rate to ion concentration).

Hence, I feel that because of the experimental results, this work should be published, and I'm happy for it to be in a high-profile journal if presented in an appropriate way. I do not feel that the work

should be rejected because I think the experiments are a significant finding that will spark a lot of new questions and research in atmospheric aerosols and climate. However, I am recommending major revisions for the authors to add appropriate discussions to the theory and necessary updates to the atmospheric calculations and discussion.

Please include line numbers in the next round of reviews, if possible, as it makes identifying text in my review easier.

#####

Major comments

#####

*** What is the composition the ions and what are their properties? ***

Understanding the species that make up the ions and their properties are also crucial towards understanding if this ion condensation will have a large impact. The ion-enhanced-condensation theory developed in the manuscript assumes that (1) the compounds included in the ions would not have condensed if they had not been part of the ion and (2) all of the ion mass condenses irreversibly, yet neither of these assumptions are explicitly discussed. Both of these assumptions depend on what species are part of the ionic molecules and clusters. Below, I describe how each of these assumptions might not be true or fully true.

#1 (the compounds included in the ions would not have condensed if they had not been part of the ion). Arnold et al. (2008, https://link.springer.com/chapter/10.1007%2F978-0-387-87664-1_14) provides an overview of what atmospheric ions are composed of, and they describe clusters of sulfuric acid and nitric acid (with one of the acids singly deprotonated) for negative clusters and H₃O⁺ clustered with small organics for positive ions. Recent papers with API-ToF measurements also show clusters containing larger, very oxidized organic molecules. (e.g. <http://www.atmos-chem-phys-discuss.net/acp-2017-481/acp-2017-481.pdf>). The sulfuric acid in the clusters will condense regardless of whether it's in ionic clusters or not. At steady state, the rate of sulfuric acid condensation will balance the chemical production of sulfuric acid regardless of what fraction of the sulfuric acid becomes part of ionic clusters and any differences in the condensation rates between the ions and neutral molecules or clusters. Thus, at steady state, ionic sulfuric acid should not be considered as additional condensing mass (above a calculation where all sulfuric acid was assumed to be neutral). It's possible that other species involved in ionic clusters (e.g. low-volatility organics) may similarly condense regardless of whether they were neutral or ionic. In the experiments in the

manuscript, it would make sense that the negative clusters were dominated by sulfuric acid (with water), but it's unclear what the positive ions might be.

#2 (all of the ionic mass condenses irreversibly). It's unclear as to whether all of the species involved in ionic clusters will condense irreversibly. Sulfuric acid certainly would and many of the larger, very oxidized organics likely would too. However, these molecules fall into #1 above: they would condense even if they were not in the ionic clusters. For the other molecules that would not have condensed anyway, is their condensation reversible? I do not know the answer to this. Does nitric acid associated with a negative ionic cluster stay in the particle after the ion condenses? Nitric acid would normally quickly re-evaporate from an acidic particle, and most nitric acid would stay in the gas phase. Does nitric acid in the cluster stay condensed to an acidic particle? What happens if you have proton transfer in the particle phase and the ion moves around between molecules? What about the small organics that are too volatile to condense irreversibly on their own?

The authors appear to include the contribution of the water mass in the mass of the ion. Should we be considering the wet or dry growth rates (dry growth rates would exclude the water uptake)? If considering the wet growth rates, is the addition of water with the ion condensing irreversibly or will the water uptake relax to the equilibrium value of a neutral particle?

These are important questions for both the experiments and for the atmosphere where the composition of the ions and aerosol acidity varies spatially and temporally.

The authors should have information on the ionic compositions from the API-ToF, yet they do not discuss these data other than for estimating the mean molecular weight of the ions. This composition information and how it relates to the two issues that I have raised above needs to be discussed in the manuscript. Also, the authors should discuss how the ions observed in the chamber compare to those expected in different parts of the atmosphere. There needs to be a clear discussion of the assumptions in the developed theory of (1) ions are material that wouldn't have otherwise condensed and (2) all of the ionic mass condenses irreversibly in the development of the theory. These assumptions should be reiterated again during the discussion section (add discussion of what if these are not fully true... they shouldn't be inasmuch as the ions are made of sulfuric acid).

*** Condensation sink of ions ***

A crucial detail that is not discussed in the manuscript, but is central to the authors' calculation, is the condensation sink of ions (usually expressed in $1/\text{time}$). The authors develop the theory of the ion-condensation correction using the concentration rate of ions (e.g. Equation 6), but then only

specify the ion formation rate in calculations (e.g. Figure 1) and the experiments and associated analyses (e.g. Figures 3 and 4). The condensation sink of the ions is the property of the existing aerosols that relates the ion formation rate to the associated steady-state ion concentration that would be used in equation 6. This condensation sink is proportional to the integral over the number distribution multiplied by the size-dependent betas.

For a given ion formation rate, air with a large ion condensation sink will have a lower steady-state ion concentration than air with a small ion condensation sink. Thus, for a given ion formation rate, the extra particle growth rate due to ions is lower when the condensation sink is higher because the ions are condensing across more particles (each particle gets fewer ions condensing). Thus, the values in Figure 1b depend on the pre-existing aerosol size distribution, yet the condensation sink used in the calculation of Figure 1b was not specified.

Similarly, in order to generalize the results from the experiment to the atmosphere (as considered on pages 11 and 12), one must consider the differences in the condensation sink between the experiments and different regions of the atmosphere (and this was not done in the calculations on pages 11 and 12). Regarding the experiments, the authors mention that they keep the aerosol concentrations low in order to ignore the effects of coagulation. This would imply that the condensation sink in the experiments may be relatively low compared to many atmospheric conditions, which would yield a higher steady-state ion concentration in the experiments for any given ion formation rate. However, this connection was not considered when calculating the atmospheric effects on pages 11 and 12.

The condensation-sink timescale (the inverse of the condensation sink) is also relevant in the interpretation of the experiments as it is the timescale for reaching the steady-state ion concentrations. For example, if the condensation sink is low in the chamber (low aerosol concentration), and the condensation sink timescale is on the order of e.g. 1 hour, it would take on the order of an hour for the ion concentrations to adjust towards a new steady-state concentration after the gamma radiation has changed. Thus, under these low-condensation-sink conditions, one would not expect a perfect step change in the growth rates. In Figure 3, the red and blue lines are straight with a fixed slope, which appears to show that the authors assumed that the ion concentrations changed to the new steady-state value instantly when the radiation changed. It's not clear that this is a good assumption (depends on the condensation-sink timescale).

The authors must consider the role of the condensation sink in their calculations, experimental analysis, and atmospheric interpretation. They need to explicitly state the condensation sink values used in their calculations.

*** Issues with calculation of atmospheric implications ***

Equation 9: This equation is not correct. Kuang et al. (2009), the paper cited for eqn 9, have a somewhat different equation that uses N_{3nm} (the number of 3nm particles at the start of growth) and N_t , the number of particles at whatever size the mode has grown to at time t . These are very different values than N_{cn} (the total number of particles larger than e.g. 3nm) and N_{ccn} (the total number of particles larger than e.g. 100nm). The Kuang et al. (2009) equation cannot be directly applied to N_{cn} and N_{ccn} since N_{cn} and N_{ccn} are integral quantities rather than the number of particles at a given size (e.g. ccn will continue to be lost as grow beyond 100 nm).

Application of Equation 9 to the atmosphere: In addition to Equation 9 being incorrect, the application of it to the atmosphere in the manuscript also makes the assumption that all CCN are formed by nucleation and growth; however many (Merikanto et al. 2009 estimates half) are from primary emissions. Ultrafine primary emissions may benefit from having extra condensing material (though not by as much as the nucleated particles) and many primary CCN started at sizes where they already acted as CCN. The additional condensible material through ions will thus impact CCN by less than this calculation predicts (maybe by about a factor of 2).

*** Abstract ***

The abstract of the paper focuses almost entirely on the results of the back-of-envelope calculations of the atmospheric implications of condensation of atmospheric ions rather than the experimental results or theory. Given the major issues that I have highlighted above, I am in no way comfortable with these atmospheric numbers being the centerpiece of the abstract (there are too many factors not considered and issues with the calculations to have faith in these numbers). The abstract should be rooted in portions of the paper that have better confidence, such as the experimental results or results of the theory with necessary assumptions stated. In my opinion, the experimental results, particularly Figure 3b is the most groundbreaking and solid part of the work. It would make sense to make these results the foundation of the paper and abstract.

#####

Specific comments

#####

Top of page 3: “Changing the ionization is therefore not expected to have an influence on the number of CCN through Coulomb interactions between aerosols.” There is a possible exception to this. If nucleation occurs through ion-induced mechanisms, the recently formed particles will be more charged than what equilibrium would predict, increasing the growth rates (Nadykto, A. B. and Yu, F.: Uptake of neutral polar vapor molecules by charged clusters/ particles: Enhancement due to dipole-charge interaction, *J. Geophys. Res.*, 108, 4717, doi:10.1029/2003JD003664, 2003.). If cosmic rays modulate the fraction of particles forming through ion-induced nucleation, then cosmic rays may affect the growth rates of these smallest particles. Snow-Kropla et al. (2011) explored this and showed it increased the impact of cosmic rays on CCN relative to assuming cosmic rays only impacted nucleation rates. My intuition is that this effect cannot explain the experimental results though.

Page 5: How good is the approximation of symmetry between ions (same mass, interaction coef etc)? The API-ToF should have this information. Is Mion/Mo similar for both positive and negative ions (relates point made earlier about providing more information on what ions are from the API-ToF).

Figure 1: The caption says the sulfuric acid concentration was $1E6 \text{ cm}^{-3}$ yet the text on page 6 says it's $5E6 \text{ cm}^{-3}$. Which one is it?

Figures 1 and 5: Where do the values (and the theory of the calculation) for the betas come from? Is it citation 20?

Figure 2 caption. I did not understand the sentence about the Gaussian fit until I read the section in the methods and saw Figure 8. I suggest removing the sentence in the Figure 2 caption or revising it.

Figure 4: I'm confused by the math here: $71-16 = 55$ (not 45) and $212-16=196$ (not 186).

Page 9: “deltaT is increasing”, do you mean “deltaT is positive”. If you do mean “increase”, what does this mean here?

Page 10: “Therefore the relative effect on the growth caused by the ions was more than an order of magnitude smaller, as can be seen from Eq. (7).” But you also had a very low condensation sink right (in order to keep coagulation negligible), right? This makes the steady state concentration of ions potentially higher in the chamber relative to the atmosphere. Was the production rate of sulfuric acid higher than expected in the atmosphere (if the condensation sink for ions is low, the

condensation sink for sulfuric acid will also be low, so maybe the production of sulfuric acid is not far off from the atmosphere even though the concentration is due to the low condensation sink).

Figures 3 and 4 and throughout. The values of DeltaT would be more meaningful if we knew what t1 (or t2) was. How large a fraction change is this (I guess this is what on the right hand side of figure 4)?

Page 11: "The present range of ion production in the atmosphere is 2-35 ions-pairs s-1 cm-3 and Fig. 1b show that 5 - 20% of the growth by condensation is caused by ions. A 20% Solar cycle variation in the ion production impacts the growth rate by 1-4%." But what condensation sink for ions does Figure 1b assume (in order to calculate the steady-state ion concentrations from the ion-pair formation rates to use in Equation 6), and how relevant is this condensation-sink value in the different parts of the atmosphere that would have these different ion formation rates?

Methods, Figure 5: The caption says that the betas were calculated assuming a MW of 100 AMU. Was 100 AMU assumed for the betas for the ions in the calculations. This contradicts the Mion/Mo of 2.25, which would put Mion at around 220 AMU. This biases the contribution of ions to condensation high as a lower Mion gives a higher beta_ion, yet a the high Mion on the Mion/Mo ratio favors more condensation per ion. This Mion may be inconsistent here with the choice of Mion benefiting the effect of ionic condensation in each portion. Please be consistent using the same Mion in both portions of the calculation.

Table II and associated discussion: I question the inclusion of the water mass in the calculation of Mion. Are we considering dry or wet growth? Is the water condensation in the ions irreversible?

#####

Grammar/spelling issues

#####

Page 2: "to be *too* small to affect clouds"

Bottom of page 2: "which *lowers* the probability for the growing aerosol to *be* lost*,* and more aerosols can survive to CCN sizes."

Page 3 : "as a result, the *charged* ion density"

Page 10: "charged ion*s* and aerosol"

Page 11: "atmospheric*ly* relevant concentrations of SA."

"AUT*H*OR CONTRIBUTIONS"

Reviewer #2 (Remarks to the Author):

Review of: "The role of ions in the growth of aerosols into cloud condensation nuclei" by Svensmark et al.

This paper presents theoretical derivation and lab experiments, demonstrating the importance of the mass flux associated with ions to the aerosol grow rate in the atmosphere. They demonstrate that the mass associated with the aerosol charging by ions and ion-aerosol recombination is important and should be considered.

General comments:

1. The processes suggested in this paper are indeed interesting although the link to clouds is too simplistic. There is no uniform CCN properties and CCN effect on clouds. The effect on clouds has a strong regime dependence. It depends on the type of clouds and the environmental conditions (as well as the environmental aerosol loading). Adding CCNs in aerosol-diluted environment can create the opposite effect as compared to adding CCNs in polluted one. The radiative forcing of CCN enhanced shallow clouds can be completely different than high cloud (deep convection, alto or cirrus clouds).
2. Similarly, when discussing aerosol cloud interactions and the overall climatic impact – the expected effects should be addressed with caution. For example, aerosol activation depends on the aerosol concentration, size distribution, and on the cloud's supersaturation (S) which depends again on the type of cloud and the environment. Larger updrafts can yield larger S and therefore activation of smaller particles. It is true that an increase of the aerosol size will make them easier to be

activated but it will not always be the limiting factor for cloud development. For example, in places where the concentration is relatively low the clouds S might be enough to activate the aerosols anyway and on the other hand in polluted environment, there can be enough large CCN's without this effect.

3. On the same note, larger aerosols can be CCNs as long as they can interact with the clouds. This defines a limited volume of the atmosphere near cloud bases in which aerosols interact with clouds and effect their microphysics. The authors again, keep the description very general. They should explain where along the vertical profile, the effect on the aerosol size by the proposed ion-particle interactions is important. Is it homogenies throughout the troposphere? Or is it likely to be more important higher in the atmosphere where the ion concentration is higher?

More specific comments:

1. "...cloud condensation nuclei (CCN) of size 50-100 nm". Why do you cut the CCN size distribution from above at 100nm? Aerosol as large as 1 μm and even 10 μm are available in the atmosphere and would act as very efficient CCNs.

2. Do you have any reference that support the claim that aerosol of size of 50nm are efficient as CCNs?

3. "... and changes in the number of CCN will influence the cloud microphysics". See my comment above and add references to support your statements.

4. "...influencing the density of CCN in the atmosphere and thereby Earth's cloud cover". Again, please see the above comments. The link between CCN concentration and cloud cover is not as simple.

5. "This conclusion stands in stark contrast to a recent experiment demonstrating that when excess ions are present in the experimental volume, all extra nucleated aerosols can grow to CCN sizes." If it is based on previous papers please provide references.

6. "Cosmic rays are the main producers of ions in Earth's lower atmosphere." Please give reference. Is it true also for the boundary layer (lower ~ 1 km) where the effect of aerosols as CCN is most important? What about radon and other terrestrial sources. I think that near the surface they are important and therefore their ions are likely to interact with aerosols and to form CCN's.

7. "...the increased recombination ensures that the equilibrium aerosol charged fraction remains the same" Please give reference.

8. "... each time an ion recombines with or charges an aerosol, a small mass (mion) is added to the aerosol.". I understand that recombination ends with the ion mass being added to the aerosol, but why ionization of the aerosol without recombination increases the mass of the aerosol?

9. "This approximation is valid for aerosols larger than about 4 nm and growth rates no more than a few nm/hour." But in your case you start with ~ 1.7 nm size aerosols. What would be the difference in this case?

10. Fig 1b: is the range of ionization rates presented here is suitable for the lower atmosphere?
11. "The ionization in the chamber could be varied from 16 to 212 ion-pairs $\text{cm}^{-3} \text{s}^{-1}$ using two γ -sources." How these ionization rates are compare to lower atmospheric conditions?
12. "One important feature is that the effect on the growth rate continues up to ~ 20 nm as can be seen in Fig. 3". It is still too small for being CCN. Would your suggest mechanism be important in bringing the aerosol to CCN size?
13. "The present range of ion production in the atmosphere is 2-35 ions-pairs $\text{s}^{-1} \text{cm}^{-3}$ " at which height? It is significantly lower than the ionization rates used in the experiment.
14. "Over the Solar cycle Γ changes by 1-4%, giving a δ_{ccn} of 0.8-3.2%." would an order of $\sim 1\%$ change in CCN concentration could effect the cloud properties?

Reviewer #3 (Remarks to the Author):

The role of ion condensation in the growth of nanometer particles into cloud condensation nuclei (CCN) has been investigated both theoretically and experimentally. The authors conclude that the mass of small ions added continuously to nucleation mode particles through ion condensation may enhance particle growth rates (by up to ~ 5 -20% under present day atmospheric conditions) and hence CCN concentrations. The enhancement is proportional to ion density and the authors propose that this ion condensation mechanism can provide a physical link between cosmic rays and clouds.

The idea of ion condensation is novel and interesting. The theoretical derivations appear sound to me. The total amount of experimental runs (3100 hours) is also impressive. The good agreement of theoretical predictions with experimental measurements indicates that the proposed mechanism could indeed be important. My main concern is the uncertainty in the estimated magnitude of CCN change associated with ion condensation and implication to clouds. I recommend the publication of this manuscript after the following comments are properly addressed.

Main comments:

1. As the authors have emphasized, the effect of ion condensation depends strongly on condensable gas concentration (n_0 in equation 6, also see Fig. 4). The conclusion of ~ 5 -20% of growth enhancement due to ion condensation was derived from Fig. 1b (see page 11, second paragraph)

and was based on $n_0=1E6 \text{ cm}^{-3}$. As I understand, n_0 should be much larger in the troposphere where significant new particle formation and growth occurs. The authors cited a reference to state that “the globally averaged time for a nucleated aerosol to grow to CCN sizes is typically 7 days”. However, numerous measurements of particle size distributions indicate that nucleated particles can grow to CCN within half day. It is inappropriate to use global averaged n_0 to estimate the magnitude of ion condensation enhancement for the following reasons:

(1) Observations indicate that most of nucleation and growth occur during the periods (usually daytime) and in regions where n_0 is generally $> \sim 5E6/\text{cm}^3$.

(2) For particle growth, sulfuric acid (SA) is not the only precursor. It has been well recognized that low volatile organics can dominate the growth rates in many locations. Nitrate and ammonium also contribute to the growth. So it is necessary to take into account species other than SA for the value of n_0 .

(3) Likely the value of m_{ion}/m_0 also depends on n_0 . $m_{\text{ion}}/m_0 = 2.25$ used in the theoretical calculation was derived from experimental data when $n_0 > \sim 2E7 \text{ cm}^{-3}$. If $n_0=1E6 \text{ cm}^{-3}$, I think that m_{ion}/m_0 will be much less than 2.25. On the other hand, m_{ion}/m_0 can be larger than 2.25 in the atmosphere if organics are considered.

2. The mass of ions comes from the clustering of ions with low volatile precursors (SA, organics). If ion condensation becomes significant (i.e., 50%), n_0 will be reduced and thus the overall growth rate (ion condensation + n_0 condensation) will be reduced. This may cancel some of the effect of ion condensation.

Minor comments:

1. How was n_{ion} calculated?

2. Figure 3: y-axis should be ΔT instead of “T”?

3. Page 2, Second line from the bottom: “lovers” should be “lowers”?

Dear reviewers,

Thank you for all your comments. They are both thoughtful and constructive, which we really appreciate. Below we reply point by point, with our replies in italicized text. Line numbers have been added to the revised manuscript and we use these to refer to changes. Furthermore new or changed text is shown in red in the revised manuscript.

CCN

Since all three reviewers have commented on the CCN calculations we collect our reply to this issue in a shared section here. Our original idea with the CCN calculation and discussion was to make a simplified attempt to gauge the potential atmospheric effect of the proposed mechanism. As the reviewers rightfully have pointed out the atmosphere is not a simple place and a more complex modelling work is needed to properly estimate the impact of the mechanism. Therefore we have removed the calculations (eq. 9, 10, 11 in the original manuscript) and instead we've added a discussion (starting LINE 264) of the possible relevance of the mechanism. The main point in the new section is that clean air seems optimal for the mechanism and that there is some observational evidence to support that there could be a global impact, which warrants further and much more detailed investigation.

Reviewers' comments:

Reviewer #1 (Remarks to the Author):

Major comments
#####

*** What is the composition the ions and what are their properties? ***

Understanding the species that make up the ions and their properties are also crucial towards understanding if this ion condensation will have a large impact. The ion-enhanced-condensation theory developed in the manuscript assumes that (1) the compounds included in the ions would not have condensed if they had not been part of the ion and (2) all of the ion mass condenses irreversibly, yet neither of these assumptions are explicitly discussed. Both of these assumptions depend on what species are part of the ionic molecules and clusters. Below, I describe how each of these assumptions might not be true of fully true.

#1 (the compounds included in the ions would not have condensed if they had not been part of the ion). Arnold et al. (2008, https://link.springer.com/chapter/10.1007%2F978-0-387-87664-1_14) provides an overview of what atmospheric ions are composed of, and they describe clusters of sulfuric acid and nitric acid (with one of the acids singly deprotonated) for negative clusters and H₃O⁺ clustered with small organics for positive ions. Recent papers with API-ToF measurements also show clusters containing larger, very oxidized organic molecules. (e.g. <http://www.atmos-chem-phys-discuss.net/acp-2017-481/acp-2017-481.pdf>). The sulfuric acid in the clusters will condense regardless of whether it's in ionic clusters or not. At steady state, the rate of sulfuric acid condensation will balance the chemical production of sulfuric acid regardless of what fraction of the

sulfuric acid becomes part of ionic clusters and any differences in the condensation rates between the ions and neutral molecules or clusters. Thus, at steady state, ionic sulfuric acid should not be considered as additional condensing mass (above a calculation where all sulfuric acid was assumed to be neutral). It's possible that other species involved in ionic clusters (e.g. low-volatility organics) may similarly condense regardless of whether they were neutral or ionic. In the experiments in the manuscript, it would make sense that the negative clusters were dominated by sulfuric acid (with water), but it's unclear what the positive ions might be.

The negative ions are mainly sulphuric acid clusters (based on API-ToF measurements), but unfortunately we have no measurements of the positive ions. A discussion of the issue of whether the compounds would have condensed anyway in steady state has been added to the manuscript (LINE 219). Based on eq. 5 and 6 we argue that while taking this into consideration does decrease the effect of the mechanism slightly this adjustment is an order of magnitude lower than the increase in growth rate from the mechanism. It is true that in steady state the condensation will balance the production but there rate is also important since it grows the particles out of the smallest sizes where they are the most vulnerable to scavenging.

#2 (all of the ionic mass condenses irreversibly). It's unclear as to whether all of the species involved in ionic clusters will condense irreversibly. Sulfuric acid certainly would and many of the larger, very oxidized organics likely would too. However, these molecules fall into #1 above: they would condense even if they were not in the ionic clusters. For the other molecules that would not have condensed anyway, is their condensation reversible? I do not know the answer to this. Does nitric acid associated with a negative ionic cluster stay in the particle after the ion condenses? Nitric acid would normally quickly re-evaporate from an acidic particle, and most nitric acid would stay in the gas phase. Does nitric acid in the cluster stay condensed to an acidic particle? What happens if you have proton transfer in the particle phase and the ion moves around between molecules? What about the small organics that are too volatile to condense irreversibly on their own?

This is also a good point and somewhat more difficult to answer. We have added this to the discussion – as you say it will depend greatly on what the actual condensing material is, so depending on the atmospheric composition this can potentially decrease the effect of the mechanism significantly.

The authors appear to include the contribution of the water mass in the mass of the ion. Should we be considering the wet or dry growth rates (dry growth rates would exclude the water uptake)? If considering the wet growth rates, is the addition of water with the ion condensing irreversibly or will the water uptake relax to the equilibrium value of a neutral particle?

There is a potential issue with sulphuric acid ions typically having more water than their neutral counterpart. Does this extra water evaporate from the aerosol once the aerosol becomes neutralized? A note about this has also been added to the discussion.

These are important questions for both the experiments and for the atmosphere where the composition of the ions and aerosol acidity varies spatially and temporally.

The authors should have information on the ionic compositions from the API-ToF, yet they do not discuss these data other than for estimating the mean molecular weight of the ions. This composition information and how it relates to the two issues that I have raised above needs to be

discussed in the manuscript. Also, the authors should discuss how the ions observed in the chamber compare to those expected in different parts of the atmosphere. There needs to be a clear discussion of the assumptions in the developed theory of (1) ions are material that wouldn't have otherwise condensed and (2) all of the ionic mass condenses irreversibly in the development of the theory. These assumptions should be reiterated again during the discussion section (add discussion of what if these are not fully true... they shouldn't be inasmuch as the ions are made of sulfuric acid).

The discussion of the mentioned assumptions have all been added to the Discussion section. We've added a line to the theory section mentioning that the caveats to the theory will be examined in the Discussion (LINE 125).

*** Condensation sink of ions ***

A crucial detail that is not discussed in the manuscript, but is central to the authors' calculation, is the condensation sink of ions (usually expressed in 1/time). The authors develop the theory of the ion-condensation correction using the concentration rate of ions (e.g. Equation 6), but then only specify the ion formation rate in calculations (e.g. Figure 1) and the experiments and associated analyses (e.g. Figures 3 and 4). The condensation sink of the ions is the property of the existing aerosols that relates the ion formation rate to the associated steady-state ion concentration that would be used in equation 6. This condensation sink is proportional to the integral over the number distribution multiplied by the size-dependent betas.

For a given ion formation rate, air with a large ion condensation sink will have a lower steady-state ion concentration than air with a small ion condensation sink. Thus, for a given ion formation rate, the extra particle growth rate due to ions is lower when the condensation sink is higher because the ions are condensing across more particles (each particle gets fewer ions condensing). Thus, the values in Figure 1b depend on the pre-existing aerosol size distribution, yet the condensation sink used in the calculation of Figure 1b was not specified.

Similarly, in order to generalize the results from the experiment to the atmosphere (as considered on pages 11 and 12), one must consider the differences in the condensation sink between the experiments and different regions of the atmosphere (and this was not done in the calculations on pages 11 and 12). Regarding the experiments, the authors mention that they keep the aerosol concentrations low in order to ignore the effects of coagulation. This would imply that the condensation sink in the experiments may be relatively low compared to many atmospheric conditions, which would yield a higher steady-state ion concentration in the experiments for any given ion formation rate. However, this connection was not considered when calculating the atmospheric effects on pages 11 and 12.

The condensation-sink timescale (the inverse of the condensation sink) is also relevant in the interpretation of the experiments as it is the timescale for reaching the steady-state ion concentrations. For example, if the condensation sink is low in the chamber (low aerosol concentration), and the condensation sink timescale is on the order of e.g. 1 hour, it would take on the order of an hour for the ion concentrations to adjust towards a new steady-state concentration after the gamma radiation has changed. Thus, under these low-condensation-sink conditions, one would not expect a perfect step change in the growth rates. In Figure 3, the red and blue lines are straight with a fixed slope, which appears to show that the authors assumed that the ion

concentrations changed to the new steady-state value instantly when the radiation changed. It's not clear that this is a good assumption (depends on the condensation-sink timescale).

Regarding the timescale for the changes to ion concentration: Measurements with a Gerdien tube has shown that the ion concentration adjusts quickly to changes in ion production (timescale of a few minutes). These measurements were typically with few or no aerosols present, but still with wall losses. Wall losses for sulphuric acid has been estimated to be about $7e-4$ s⁻¹ in previous experiments – this will be slightly lower for the slightly larger ions, but not by much. For comparison the Condensation Sink was calculated for a typical size distribution during one of the experiments (exp V2) using SMPS data with about 4000 cm⁻³ aerosols present and the CS was $1.2e-4$ s⁻¹, meaning that the wall loss is dominant over the CS, so the Gerdien measurements with wall losses only are representative for what happens in the chamber during the experiments, with regards to the timescale for adjustment of the ion concentration.

The second issue with the condensation sink is to what degree it affects the actual ion concentration. For an ion production of 16 cm⁻³ s⁻¹ the ion concentration in the chamber, taking wall losses and CS into account, is 92% of what you get when using recombination and loss. Using wall losses but disregarding CS this becomes 93%. For a q of 212 cm⁻³ s⁻¹ the ion concentration is 98% of what you get with only recombination as loss, with and without CS. So depending on the atmospheric conditions this becomes a factor of >0.9 to be multiplied onto the n_{ion} in eq. 6 of the manuscript, leading to a small reduction in the calculated Gamma value, but not anything that affects the results of the paper significantly. We've added a paragraph detailing this issue to the paper (LINE 246).

The authors must consider the role of the condensation sink in their calculations, experimental analysis, and atmospheric interpretation. They need to explicitly state the condensation sink values used in their calculations.

*** Issues with calculation of atmospheric implications ***

Equation 9: This equation is not correct. Kuang et al. (2009), the paper cited for eqn 9, have a somewhat different equation that uses N_{3nm} (the number of 3nm particles at the start of growth) and N_t , the number of particles at whatever size the mode has grown to at time t . These are very different values than N_{cn} (the total number of particles larger than e.g. 3nm) and N_{ccn} (the total number of particles larger than e.g. 100nm). The Kuang et al. (2009) equation cannot be directly applied to N_{cn} and N_{ccn} since N_{cn} and N_{ccn} are integral quantities rather than the number of particles at a given size (e.g. ccn will continue to be lost as grow beyond 100 nm).

Application of Equation 9 to the atmosphere: In addition to Equation 9 being incorrect, the application of it to the atmosphere in the manuscript also makes the assumption that all CCN are formed by nucleation and growth; however many (Merikanto et al. 2009 estimates half) are from primary emissions. Ultrafine primary emissions may benefit from having extra condensing material (though not by as much as the nucleated particles) and many primary CCN started at sizes where they already acted as CCN. The additional condensible material through ions will thus impact CCN by less than this calculation predicts (maybe by about a factor of 2).

As mentioned above the entire discussion of implications has been removed in its original form and replaced with a more qualitative discussion.

*** Abstract ***

The abstract of the paper focuses almost entirely on the results of the back-of-envelope calculations of the atmospheric implications of condensation of atmospheric ions rather than the experimental results or theory. Given the major issues that I have highlighted above, I am in no way comfortable with these atmospheric numbers being the centerpiece of the abstract (there are too many factors not considered and issues with the calculations to have faith in these numbers). The abstract should be rooted in portions of the paper that have better confidence, such as the experimental results or results of the theory with necessary assumptions stated. In my opinion, the experimental results, particularly Figure 3b is the most groundbreaking and solid part of the work. It would make sense to make these results the foundation of the paper and abstract.

The abstract has been completely rewritten putting focus on the experiments and theory. Due to the limits on the length of the abstract the assumptions unfortunately cannot be stated in detail in the abstract.

Specific comments
#####

Top of page 3: “Changing the ionization is therefore not expected to have an influence on the number of CCN through Coulomb interactions between aerosols.” There is a possible exception to this. If nucleation occurs through ion-induced mechanisms, the recently formed particles will be more charged than what equilibrium would predict, increasing the growth rates (Nadykto, A. B. and Yu, F.: Uptake of neutral polar vapor molecules by charged clusters/ particles: Enhancement due to dipole-charge interaction, J. Geophys. Res., 108, 4717, doi:10.1029/2003JD003664, 2003.). If cosmic rays modulate the fraction of particles forming through ion-induced nucleation, then cosmic rays may affect the growth rates of these smallest particles. Snow-Kropla et al. (2011) explored this and showed it increased the impact of cosmic rays on CCN relative to assuming cosmic rays only impacted nucleation rates. My intuition is that this effect cannot explain the experimental results though.

It's certainly true that ion induced nucleation will produce a high overcharging of the smallest sizes and that there is an increased growth rate due to the mentioned charge interactions. The overcharging is however quite quickly neutralized, depending on the ion-concentration and growth rate. In Laakso et al, ACP 7 p 1333, 2007 the overcharge (in ambient measurements) is gone by about 5 nm and as shown in the references you mention the dipole-charge effect also diminishes quickly with size. So while there probably is a small enhancement in growth rate at the smallest sized that effect should not persist up to anywhere near the 20 nm where an effect is detected in this work. A line about this effect was added (LINE 65).

Page 5: How good is the approximation of symmetry between ions (same mass, interaction coef etc)? The API-ToF should have this information. Is Mion/Mo similar for both positive and negative ions (relates point made earlier about providing more information on what ions are from the API-ToF).

Unfortunately we do not have measurements of the positive ions. Generally positive ions have been observed to be slightly larger than negative ions (e.g. Horrak et al, JGR 103 D12, p 13909, 1998).

So the assumption is certainly not perfect. Hoppel and Frick (Aerosol Sci. and Tech, 5:1, p 1-21, 1986 use masses of 150 and 90 AMU for positive and negative ions respectively, but the exact relation in our experiments (or in the atmosphere in general) is not known. A note about this has been added (LINE 124).

Figure 1: The caption says the sulfuric acid concentration was $1E6 \text{ cm}^{-3}$ yet the text on page 6 says it's $5E6 \text{ cm}^{-3}$. Which one is it?

1e6 cm-3 is correct, this has been corrected in the manuscript.

Figures 1 and 5: Where do the values (and the theory of the calculation) for the betas come from? Is it citation 20?

The theory and calculation of the interaction coefficients is based on F. Yu and R. P. Turco, Journal of Geophysical Research 103, 25915 (1998). Which is now reference [25].

Figure 2 caption. I did not understand the sentence about the Gaussian fit until I read the section in the methods and saw Figure 8. I suggest removing the sentence in the Figure 2 caption or revising it.

Added a reference to the Methods section.

Figure 4: I'm confused by the math here: $71-16 = 55$ (not 45) and $212-16=196$ (not 186).

Good catch, the values in the figure are correct, the caption has been corrected.

Page 9: "deltaT is increasing", do you mean "deltaT is positive". If you do mean "increase", what does this mean here?

Since there is (nearly) constant difference in growth rate between the two profiles (with and without gamma sources on) the time difference between when the two profiles reaches a given size increases, so that is what we mean. We've added notes to make this more clear at LINE 164 and 166.

Page 10: "Therefore the relative effect on the growth caused by the ions was more than an order of magnitude smaller, as can be seen from Eq. (7)." But you also had a very low condensation sink right (in order to keep coagulation negligible), right? This makes the steady state concentration of ions potentially higher in the chamber relative to the atmosphere. Was the production rate of sulfuric acid higher than expected in the atmosphere (if the condensation sink for ions is low, the condensation sink for sulfuric acid will also be low, so maybe the production of sulfuric acid is not far off from the atmosphere even though the concentration is due to the low condensation sink).

The wall-loss rate of SA in the chamber has previously been estimated to be about $7e-4 \text{ s}^{-1}$. With a concentration of $2e7 \text{ cm}^{-3}$ of sulphuric acid in the chamber this implies a production rate of $\sim 1e4 \text{ cm}^{-3} \text{ s}^{-1}$, in the case of low condensation sink as we have in our experiment (see previous answer).

Figures 3 and 4 and throughout. The values of DeltaT would be more meaningful if we knew what

t1 (or t2) was. How large a fraction change is this (I guess this is what on the right hand side of figure 4)?

Yes, the fractional change is shown on the right hand axis (converted to dr/dt). Absolute values for t1 and t2 can be seen for one size in one of the experiments (V9) in fig. 8 (original manuscript).

Page 11: “The present range of ion production in the atmosphere is 2-35 ions-pairs s-1 cm-3 and Fig. 1b show that 5 - 20% of the growth by condensation is caused by ions. A 20% Solar cycle variation in the ion production impacts the growth rate by 1-4%.” But what condensation sink for ions does Figure 1b assume (in order to calculate the steady-state ion concentrations from the ion-pair formation rates to use in Equation 6), and how relevant is this condensation-sink value in the different parts of the atmosphere that would have these different ion formation rates?

Regarding the experimental condensation sink it has very little effect (as shown in a previous answer). We’ve added a line (LINE 246) mentioning that high condensation sink can reduce the effect. Overall a clean atmosphere is most favorable for the mechanism.

Methods, Figure 5: The caption says that the betas were calculated assuming a MW of 100 AMU. Was 100 AMU assumed for the betas for the ions in the calculations. This contradicts the Mion/Mo of 2.25, which would put Mion at around 220 AMU. This biases the contribution of ions to condensation high as a lower Mion gives a higher beta_ion, yet a the high Mion on the Mion/Mo ratio favors more condensation per ion. This Mion may be inconsistent here with the choice of Mion benefiting the effect of ionic condensation in each portion. Please be consistent using the same Mion in both portions of the calculation.

The figure was ment as an example: It has now been replaced with a a neutral molecule of 100 AMU and an ion of 225 AMU which is used throughout the paper. We have included a short statement on the sensitivity in the important ratio b^{+0}/b^{00} by changing the values of the neutral mass (100 – 150 AMU) and the ion masses (200-300), within reasonable bounds and it changes up with up 20%. See LINE 261.

Table II and associated discussion: I question the inclusion of the water mass in the calculation of Mion. Are we considering dry or wet growth? Is the water condensation in the ions irreversible?

If we are considering the mass added to the aerosols the total mass, including water, should be included. As mentioned in one of the replies above there certainly is a potential issue with sulphuric acid ions typically having more water than their neutral counterpart.

Grammar/spelling issues
#####

Page 2: “to be *too* small to affect clouds”

Bottom of page 2: “which *lowers* the probability for the growing aerosol to *be* lost*,* and more aerosols can survive to CCN sizes.”

Page 3 : “as a result, the **charged** ion density”
Changed to “as a result, a change in ion density”

Page 10: “charged ion*s* and aerosol”
Changed to “ions and aerosols”

Page 11: “atmospheric*ly* relevant concentrations of SA.”

“AUT*H*OR CONTRIBUTIONS”

Fixed, thank you for finding all of these!

Reviewer #2 (Remarks to the Author):

Review of: "The role of ions in the growth of aerosols into cloud condensation nuclei" by Svensmark et al.

This paper presents theoretical derivation and lab experiments, demonstrating the importance of the mass flux associated with ions to the aerosol grow rate in the atmosphere. They demonstrate that the mass associated with the aerosol charging by ions and ion-aerosol recombination is important and should be considered.

General comments:

1. The processes suggested in this paper are indeed interesting although the link to clouds is too simplistic. There is no uniform CCN properties and CCN effect on clouds. The effect on clouds has a strong regime dependence. It depends on the type of clouds and the environmental conditions (as well as the environmental aerosol loading). Adding CCNs in aerosol-diluted environment can create the opposite effect as compared to adding CCNs in polluted one. The radiative forcing of CCN enhanced shallow clouds can be completely different than high cloud (deep convection, alto or cirrus clouds).
2. Similarly, when discussing aerosol cloud interactions and the overall climatic impact – the expected effects should be addressed with caution. For example, aerosol activation depends on the aerosol concentration, size distribution, and on the cloud's supersaturation (S) which depends again on the type of cloud and the environment. Larger updrafts can yield larger S and therefore activation of smaller particles. It is true that an increase of the aerosol size will make them easier to be activated but it will not always be the limiting factor for cloud development. For example, in places where the concentration is relatively low the clouds S might be enough to activate the aerosols anyway and on the other hand in polluted environment, there can be enough large CCN's without this effect.
3. On the same note, larger aerosols can be CCNs as long as they can interact with the clouds. This defines a limited volume of the atmosphere near cloud bases in which aerosols interact with clouds and effect their microphysics. The authors again, keep the description very general. They should explain where along the vertical profile, the effect on the aerosol size by the proposed ion-particle interactions is important. Is it homogenies throughout the troposphere? Or is it likely to be more important higher in the atmosphere where the ion concentration is higher?

We agree that the issue of CCN and their effect on clouds is a complicated issue. Much of the discussion about atmospheric impact has been removed and replaced with a more qualitative discussion (LINE218), which also touches on where in the atmosphere the mechanism might be relevant. In order to gauge the actual atmospheric impact more detailed modelling, beyond the scope of this work, is required. Especially the activation of CCNs into clouds is something which would require considerably more work (and space in the paper) to do justice.

More specific comments:

1. "...cloud condensation nuclei (CCN) of size 50-100 nm". Why do you cut the CCN size distribution from above at 100nm? Aerosol as large as 1 µm and even 10 µm are available in the atmosphere and would act as very efficient CCNs.

This is meant to be the minimum required size for an aerosol to act as a CCN. The manuscript has been updated to make this clear.

2. Do you have any reference that support the claim that aerosol of size of 50nm are efficient as CCNs?

We've added a reference to Seinfeld and Pandis 2006, chap. 17, where Köhler curves for varying dry diameters of salt particles are shown – the critical super saturation for a 50 nm (NH₄)₂SO₄ aerosol is about 0.45%.

3. "... and changes in the number of CCN will influence the cloud microphysics". See my comment above and add references to support your statements.

Added a line to make it clearer that the influence on clouds will depend on local conditions and a reference to a CCN model (Pierce et al, ACP 9, 2009).

4. "...influencing the density of CCN in the atmosphere and thereby Earth's cloud cover". Again, please see the above comments. The link between CCN concentration and cloud cover is not as simple.

We agree that the link between CCN and cloud cover is not simple. The cited line is just meant to summarize that there is an existing proposal that the changes in ionization can affect aerosol formation and thus cloud cover – the idea is described in more detail in the references 3-6. This paper is focused on the formation of CCN and we do not have enough space to go into great detail about the activation of CCNs.

5. "This conclusion stands in stark contrast to a recent experiment demonstrating that when excess ions are present in the experimental volume, all extra nucleated aerosols can grow to CCN sizes." If it is based on previous papers please provide references.

The reference is Svensmark 2013, which appears further down in the text. The reference has been moved to make it more clear that this is the experiment we are talking about.

6. "Cosmic rays are the main producers of ions in Earth's lower atmosphere." Please give reference. Is it true also for the boundary layer (lower ~1 km) where the effect of aerosols as CCN is most

important? What about radon and other terrestrial sources. I think that near the surface they are important and therefore their ions are likely to interact with aerosols and to form CCN's.

Terrestrial sources are, as you say, important in the boundary layer above land. Above oceans and above the boundary layer cosmic rays are the main source of ions. We've added a reference to Laakso et al, ACP 4, 2004 to support this statement. (LINE 60)

7. "...the increased recombination ensures that the equilibrium aerosol charged fraction remains the same" Please give reference.

Added a reference to Hoppel 1985, discussing the charging state of aerosols. Also note that a line was added to elaborate on how ion induced nucleation can change the charging state. (LINE 65)

8. "... each time an ion recombines with or charges an aerosol, a small mass (mion) is added to the aerosol." I understand that recombination ends with the ion mass being added to the aerosol, but why ionization of the aerosol without recombination increases the mass of the aerosol?

We just mean that mass is added each time an ion condenses onto an aerosol, which then either leads to the aerosol being charged or neutralized. The text has been changed to reflect this. (LINE 72)

9. "This approximation is valid for aerosols larger than about 4 nm and growth rates no more than a few nm/hour." But in your case you start with ~1.7nm size aerosols. What would be the difference in this case?

The condition for the approximation to be good is the steady state condition leading to Eqs. (3) in line 105. The estimate was a non rigorous assessment, and we therefore tried to test the steady state approximation by full numerical solution of the governing equations involving the variables (N^0 , N^+ , N , n^0 , n^+ and n^-). The figure below was the result of a test where the outcome of the numerical simulation and the approximation of the theory could be tested. As can be seen from the figure there seems to be a good overall agreement. The text: "This approximation is valid for aerosols larger than about 4 nm and growth rates no more than a few nm/hour". Is removed, since the condition of steady state is already stated in LINE 104.

Fig 1: The change in the function G calculated from the theory (solid black line), compared with a full numerical simulation (red diamonds). The parameters used are $Dq = 20 - 10$ ion-pairs/cm s,

$n^0 = 5 \cdot 10^7$ molecules/cm³. It illustrates that the present theory is in good agreement with the full numerical simulation.

10. Fig 1b: is the range of ionization rates presented here suitable for the lower atmosphere?

The range goes beyond what can be found for the present lower atmosphere. Higher ionization rates are included to demonstrate what might happen during extreme events such as a nearby supernova.

11. "The ionization in the chamber could be varied from 16 to 212 ion-pairs cm⁻³ s⁻¹ using two γ -sources." How do these ionization rates compare to lower atmospheric conditions?

The high ionization rate is higher than what can be found in the lower atmosphere. Since the effect is small and hard to detect we needed high ionization. This also compensates for the fact that the neutral gas concentration in the chamber is somewhat higher than found in the average atmosphere, which reduces the magnitude of the effect.

12. "One important feature is that the effect on the growth rate continues up to ~ 20 nm as can be seen in Fig. 3". It is still too small for being CCN. Would your suggested mechanism be important in bringing the aerosol to CCN size?

The early stages of growth are very important since the smallest aerosols are the most vulnerable to scavenging by large pre-existing aerosols. So even if the effect did not go beyond 20 nm it would still make a difference. If the effect does go to 20 nm we do think that it will also be effective beyond that size.

13. "The present range of ion production in the atmosphere is 2-35 ions-pairs s⁻¹ cm⁻³" at which height? It is significantly lower than the ionization rates used in the experiment.

Yes. See our reply to your comment no. 11.

14. "Over the Solar cycle Γ changes by 1-4%, giving a δ_{ccn} of 0.8-3.2%." would an order of ~1% change in CCN concentration could effect the cloud properties?

This will depend on local conditions and has to be studied in more detail. Under ideal circumstances a change in CCN could translate to an equal change in cloud cover. Note that during Forbush decreases, as described in the new discussion section, the change in cloud cover is also just a few percent.

Reviewer #3 (Remarks to the Author):

The role of ion condensation in the growth of nanometer particles into cloud condensation nuclei

(CCN) has been investigated both theoretically and experimentally. The authors conclude that the mass of small ions added continuously to nucleation mode particles through ion condensation may enhance particle growth rates (by up to ~ 5-20% under present day atmospheric conditions) and hence CCN concentrations. The enhancement is proportional to ion density and the authors propose that this ion condensation mechanism can provide a physical link between cosmic rays and clouds.

The idea of ion condensation is novel and interesting. The theoretical derivations appear sound to me. The total amount of experimental runs (3100 hours) is also impressive. The good agreement of theoretical predictions with experimental measurements indicates that the proposed mechanism could indeed be important. My main concern is the uncertainty in the estimated magnitude of CCN change associated with ion condensation and implication to clouds. I recommend the publication of this manuscript after the following comments are properly addressed.

Main comments:

1. As the authors have emphasized, the effect of ion condensation depends strongly on condensable gas concentration (n_0 in equation 6, also see Fig. 4). The conclusion of ~ 5-20% of growth enhancement due to ion condensation was derived from Fig. 1b (see page 11, second paragraph) and was based on $n_0=1E6$ cm⁻³. As I understand, n_0 should be much larger in the troposphere where significant new particle formation and growth occurs. The authors cited a reference to state that “the globally averaged time for a nucleated aerosol to grow to CCN sizes is typically 7 days”. However, numerous measurements of particle size distributions indicate that nucleated particles can grow to CCN within half day. It is inappropriate to use global averaged n_0 to estimate the magnitude of ion condensation enhancement for the following reasons:

We've removed the section about atmospheric implications and replaced it with a more qualitative discussion. There we present a few references that argue in favor of the slow growth rates. There are surely scenarios where the growth rate is much faster than what we indicate. The presented mechanism will work best at low n_0 , such as in a clean marine atmosphere.

(1) Observations indicate that most of nucleation and growth occur during the periods (usually daytime) and in regions where n_0 is generally $> \sim 5E6$ /cm³.
(2) For particle growth, sulfuric acid (SA) is not the only precursor. It has been well recognized that low volatile organics can dominate the growth rates in many locations. Nitrate and ammonium also contribute to the growth. So it is necessary to take into account species other than SA for the value of n_0 .

*This is true. In the experiments sulphuric acid is certainly the main growth agent and this will also be the case for the clean marine atmosphere, where the mechanism probably is the most effective. We've added a line to the manuscript at **LINE 275**) that other gasses may need to be taken into account for n_0 , depending on the atmospheric composition.*

(3) Likely the value of m_{ion}/m_0 also depends on n_0 . $m_{ion}/m_0 = 2.25$ used in the theoretical calculation was derived from experimental data when $n_0 > \sim 2E7$ cm⁻³. If $n_0=1E6$ cm⁻³, I think that m_{ion}/m_0 will be much less than 2.25. On the other hand, m_{ion}/m_0 can be larger than 2.25 in the atmosphere if organics are considered.

The ion mass will certainly depend on the atmospheric conditions. In Table II of the methods section we show how the negative ion mass changes with our UV settings and it appears that for

our experiment the change are rather small. In the atmosphere the variations can be expected to be larger.

2. The mass of ions comes from the clustering of ions with low volatile precursors (SA, organics). If ion condensation becomes significant (i.e., 50%), n_0 will be reduced and thus the overall growth rate (ion condensation + n_0 condensation) will be reduced. This may cancel some of the effect of ion condensation.

This is correct and we've added some discussion of this topic to the text (LINE 218), which results in a small correction to the effect calculated by eq. 6.

Minor comments:

1. How was n_{ion} calculated?

*n_{ion} was calculated simply as the square root of the ion production divided by the recombination coefficient. Taking wall losses and/or condensation sink to aerosols into account leads to a small adjustment of the ion concentration, which again leads to a minor decrease in the Gamma value found by eq. 6 in the manuscript (see reply below to Rev. 1 for details). We have also included the sentence “*Secondly, the number density of aerosols should also be small so the majority of ions are not located on aerosols*” in the discussion (LINE 267).*

2. Figure 3: y-axis should be delta T instead of “T”?

Fixed, thanks for finding this.

3. Page 2, Second line from the bottom: “lovers” should be “lowers”?

Fixed.

Reviewer #1 (Remarks to the Author):

Jeff Pierce

The authors have adequately addressed all of my minor comments and all but 1 of my major comments. I am in favor of publication once this remaining issue is adequately addressed.

Major comments addressed well

Thanks for clarifying that the recombination of small ions is the dominant sink of ions rather than condensation to particles. The jump between ion-pair formation rate and ion concentration makes sense then. It would be useful to state in the text that Figure 1 assumes that ion recombination is the only ion sink and discuss that this should be generally valid in the atmosphere in all but very polluted conditions.

The response and additions regarding the reversibility of the ionic condensation is sufficient.

I appreciate the reframing of the discussion and the abstract.

***Remaining major issue** *

I disagree with the logic the response about H₂SO₄ and HSO₄⁻ condensation and the first caveat at L219. If all of the negative ions are HSO₄⁻, there will be no enhancement of the condensation rate (at steady state) due to these negative ions. This is because all H₂SO₄ that is produced condenses whether it goes to HSO₄⁻ or not. $Cond_H_2SO_4 = Production_H_2SO_4$. Ion concentrations will not affect the condensation rates of H₂SO₄ in the atmosphere. In their response, the authors state, "It is true that in steady state the condensation will balance the production but there rate is also important since it grows the particles out of the smallest sizes where they are the most vulnerable to scavenging." However, once at steady state, the *rate* of H₂SO₄ + HSO₄⁻ condensation will be identical (if the condensation sink increases due to more of the H₂SO₄ being HSO₄⁻, the concentrations will drop and the rate will stay the same).

Are the authors accounting for the decrease in $([\text{H}_2\text{SO}_4] + [\text{HSO}_4^-])$ when the $[\text{HSO}_4^-]/[\text{H}_2\text{SO}_4]$ ratio increases (for a fixed production rate of H_2SO_4)? This is why the condensation *rate* of $([\text{H}_2\text{SO}_4] + [\text{HSO}_4^-])$ does not change when $[\text{HSO}_4^-]/[\text{H}_2\text{SO}_4]$ is modulated by a change in ions.

It might be possible that there is some effect of the adjustment time to get to steady state, but this can both increase or decrease the condensation rate relative to the steady-state assumption (based on whether the production rate went up or down, the condensation sink went up or down, or the ion formation rate went up or down). Thus, I don't see how ions affect the growth of the "particles out of the smallest sizes where they are the most vulnerable to scavenging." There should be, on average, no change in the mass flux of sulfuric acid to the particles due to ions.

Reviewer #2 (Remarks to the Author):

The authors have answered most of my specific comments. They did not give much attention to the general comments that aimed in adding a perspective on the expected effects on clouds in nature.

The described mechanism for aerosol growth is interesting and I recommend the acceptance of this paper.

However, I encourage the authors to use my previous general comments in order to put this mechanism in a climate perspective. I recommend extending their discussion on the possible impact of their finding on clouds. They discuss the possible importance to CCN budget but do not present much of the uncertainties regarding the magnitude of the described effect and its location in the atmospheric column.

They should not (and cannot) state what would be the overall impact but they should introduce some of the complexities associate with their effect. They should acknowledge that there is no one aerosol effect on clouds but instead it is regime dependent. Aerosol effects depends on the cloud type and on the background aerosol conditions. In some cases the overall effect can be an increase in the cloud coverage and in others decrease. Climatic effects on shallow marine stratocumulus are different than the effect deep convection or on polar clouds.

Reviewer #3 (Remarks to the Author):

The authors have addressed most of my concerns. My main remaining concern is with regard to the significance of the ionization on particle growth (to CCN) in the real atmosphere. In the abstract, the authors stated that "With a neutral gas concentration of $1E6$ molecules cm^{-3} and an ion density of $1E3$ ions cm^{-3} the growth from ions can constitute several percent of the neutral growth." The authors further pointed out in the text that the presented mechanism will work best at low condensing gas concentration, such as in a clean marine atmosphere. In lines 273-275, the authors cited a reference to argue that the typical growth rate of aerosols in clean marine atmosphere was estimated to be of the order of 0.4 nm/hr and said this growth rate imply an average low gas concentration of condensing gas of $n_0 \sim 1E6$ molecules cm^{-3} . How did the authors obtain $1E6$ molecules cm^{-3} (of H_2SO_4) for a growth rate of 0.4 nm/hr? Based on my calculation, 0.4 nm/hr should need H_2SO_4 gas concentration of around $5E6$ molecules cm^{-3} . If n_0 is indeed $5E6$ molecules cm^{-3} , will the effect still be potentially important in the atmosphere?

Dear Editor,

Thank you all for your work, which really improved the manuscript.

We have now addressed the remaining issues the referees raised. Our corrections can be seen in red in the resubmitted manuscript.

Reviewer #1 (Jeff Pierce)

***Remaining major issue** *

I disagree with the logic the response about H₂SO₄ and HSO₄⁻ condensation and the first caveat at L219. If all of the negative ions are HSO₄⁻, there will be no enhancement of the condensation rate (at steady state) due to these negative ions. This is because all H₂SO₄ that is produced condenses whether it goes to HSO₄⁻ or not. $\text{Cond_H}_2\text{SO}_4 = \text{Production_H}_2\text{SO}_4$. Ion concentrations will not affect the condensation rates of H₂SO₄ in the atmosphere. In their response, the authors state, "It is true that in steady state the condensation will balance the production but there rate is also important since it grows the particles out of the smallest sizes where they are the most vulnerable to scavenging." However, once at steady state, the *rate* of H₂SO₄ + HSO₄⁻ condensation will be identical (if the condensation sink increases due to more of the H₂SO₄ being HSO₄⁻, the concentrations will drop and the rate will stay the same).

Are the authors accounting for the decrease in ([H₂SO₄]+[HSO₄⁻]) when the [HSO₄⁻]/[H₂SO₄] ratio increases (for a fixed production rate of H₂SO₄)? This is why the condensation *rate* of ([H₂SO₄]+[HSO₄⁻]) does not change when [HSO₄⁻]/[H₂SO₄] is modulated by a change in ions.

It might be possible that there is some effect of the adjustment time to get to steady state, but this can both increase or decrease the condensation rate relative to the steady-state assumption (based on whether the production rate went up or down, the condensation sink went up or down, or the ion formation rate went up or down). Thus, I don't see how ions affect the growth of the "particles out of the smallest sizes where they are the most vulnerable to scavenging." There should be, on average, no change in the mass flux of sulfuric acid to the particles due to ions.

We took the liberty to contact Jeff Pierce to discuss in more detail what we could do, and the outcome was that there was a slight misunderstanding of the physics in the text. We agreed on a simple addition to avoid this. Therefore, on line 232 we have added:

"In fact, even if the larger particles grow slightly slower due to a decrease in neutral molecules, the growth rate of the smaller particles is enhanced due to the ion-interactions, which make the cross-section of the small particles larger (e.g., see Fig. 5)."

Reviewer #2 (Remarks to the Author):

The authors have answered most of my specific comments. They did not give much attention to the general comments that aimed in adding a perspective on the expected effects on clouds in nature.

The described mechanism for aerosol growth is interesting and I recommend the acceptance of this paper.

However, I encourage the authors to use my previous general comments in order to put this mechanism in a climate perspective. I recommend extending their discussion on the possible impact of their finding on clouds. They discuss the possible importance to CCN budget but do not present much of the uncertainties regarding the magnitude of the described effect and its location in the atmospheric column.

They should not (and cannot) state what would be the overall impact but they should introduce some of the complexities associate with their effect. They should acknowledge that there is no one aerosol effect on clouds but instead it is regime dependent. Aerosol effects depends on the cloud type and on the background aerosol conditions. In some cases the overall effect can be an increase in the cloud coverage and in others decrease. Climatic effects on shallow marine stratocumulus are different than the effect deep convection or on polar clouds.

On line 314 we have included the following to take into account the referee's comments:

“It should be stressed that there is not just one effect of CCN on clouds, but that the impact will depend on regional differences and cloud types. In regions with a relative high number of CCN the presented effect will small, in addition the effect on convective clouds and on ice clouds is expected to be negligible. Additional CCNs can even result in fewer clouds [39]. Since the ion-condensation effect is largest for low SA concentrations and aerosol densities the impact is believed to be largest in marine stratus clouds.”

Reviewer #3 (Remarks to the Author):

The authors have addressed most of my concerns. My main remaining concern is with regard to the significance of the ionization on particle growth (to CCN) in the real atmosphere. In the abstract, the authors stated that “With a neutral gas concentration of $1E6$ molecules cm^{-3} and an ion density of $1E3$ ions cm^{-3} the growth from ions can constitute several percent of the neutral growth.” The authors further pointed out in the text that the presented mechanism will work best at low condensing gas concentration, such as in a clean marine atmosphere. In lines 273-275, the authors cited a reference to argue that the typical growth rate of aerosols in clean marine atmosphere was estimated to be of the order of 0.4 nm/hr and said this growth rate imply an average low gas concentration of condensing gas of $n_0 \sim 1E6$ molecules cm^{-3} . How did the authors obtain $1E6$ molecules cm^{-3} (of H_2SO_4) for a growth rate of 0.4 nm/hr? Based on my calculation,

0.4 nm/hr should need H₂SO₄ gas concentration of around 5E6 molecules cm⁻³. If n₀ is indeed 5E6 molecules cm⁻³, will the effect still be potentially important in the atmosphere?

We have on line 275

“In these field measurements, the typical growth rate of aerosols was estimated to be of the order 0.4 nm/hour [36], which implies an average low gas concentration of condensing gas of $n_0 \sim 10^6$ molecules cm⁻³. Measurements and simulations of SA concentration in the free troposphere annually averaged over day and night is of the order $n_0 \sim 10^6$ molecules cm⁻³ [37]. This may well be consistent with the above slightly larger estimate, since the aerosol cross-section for scavenging smaller aerosols increases with size, which adds to the growth rate. Secondly, the observations suggest that as the aerosols enter the marine boundary layer, some of the aerosols are further grown to CCN sizes [36] [Supp. mat.]. Since the effect of ion-condensation scales inversely with n_0 , a concentration of $n_0 \sim 4 \cdot 10^6$ molecules cm⁻³ would diminish the effect by a factor of four. As can be seen in Fig. 1b, the effect of ion-condensation for an ionization rate of $q = 10$ ion-pairs cm⁻³s⁻¹ would change from 10% to 2.5% which may still be important.”

The added reference is to: Dunne *et al.* Science 354, 1119 (2016).

There in figure S10 (supplementary material) an estimate of the annual mean sulfuric acid concentration is shown, giving support to the low concentration mentioned in the present paper.

Sincerely yours,
Henrik Svensmark

Reviewer #1 (Remarks to the Author):

I recommend publication. Thanks for the edits.

Reviewer #3 (Remarks to the Author):

The authors have addressed my concern by changing the $n_0=10^6 \text{ cm}^{-3}$ to $4 \times 10^6 \text{ cm}^{-3}$ on line 277 of the revised manuscript and provided some additional discussion. I assume that " $n_0 \sim 10^6 \text{ molecules cm}^{-3}$ " given in the reply to my comment is a typo (should be $4 \times 10^6 \text{ cm}^{-3}$, as indicated in the text).

I recommend the publication of this manuscript in Nature Communications.